# A xanthene derivative, DS20060511, attenuates glucose intolerance by inducing skeletal muscle-specific GLUT4 translocation in mice

Shinji Furuzono[1,11], Tetsuya Kubota[2,3,4,5,6,7,11], Junki Taura[1,2,11], Masahiro Konishi[1], Asuka Naito[8], Masato Tsutsui[9], Hiroshi Karasawa [1✉], Naoto Kubota [2,3,4✉] & Takashi Kadowaki [2,10✉]

Reduced glucose uptake into the skeletal muscle is an important pathophysiological abnormality in type 2 diabetes, and is caused by impaired translocation of glucose transporter 4 (GLUT4) to the skeletal muscle cell surface. Here, we show a xanthene derivative, DS20060511, induces GLUT4 translocation to the skeletal muscle cell surface, thereby stimulating glucose uptake into the tissue. DS20060511 induced GLUT4 translocation and stimulated glucose uptake into differentiated L6-myotubes and into the skeletal muscles in mice. These effects were completely abolished in GLUT4 knockout mice. Induction of GLUT4 translocation by DS20060511 was independent of the insulin signaling pathways including IRS1-Akt-AS160 phosphorylation and IRS1-Rac1-actin polymerization, eNOS pathway, and AMPK pathway. Acute and chronic DS20060511 treatment attenuated the glucose intolerance in obese diabetic mice. Taken together, DS20060511 acts as a skeletal muscle-specific GLUT4 translocation enhancer to facilitate glucose uptake. Further studies of DS20060511 may pave the way for the development of novel antidiabetic medicines.

[1] End-Organ Disease Laboratories, Daiichi Sankyo Co., Ltd., Tokyo, Japan. [2] Department of Diabetes and Metabolic Diseases, Graduate School of Medicine, The University of Tokyo, Tokyo, Japan. [3] Department of Clinical Nutrition, National Institutes of Biomedical Innovation, Health and Nutrition (NIBIOHN), Tokyo, Japan. [4] Laboratory for Intestinal Ecosystem, RIKEN Center for Integrative Medical Sciences (IMS), Yokohama, Japan. [5] Division of Diabetes and Metabolism, The Institute for Medical Science, Asahi Life Foundation, Tokyo, Japan. [6] Intestinal Microbiota Project, Kanagawa Institute of Industrial Science and Technology, Ebina, Japan. [7] Division of Cardiovascular Medicine, Toho University Ohashi Medical Center, Tokyo, Japan. [8] Discovery Science and Technology Department, Daiichi Sankyo RD Novare Co., Ltd., Tokyo, Japan. [9] Department of Pharmacology, Graduate School of Medicine, University of the Ryukyus, Nishihara, Japan. [10] Toranomon Hospital, Tokyo, Japan. [11] These authors contributed equally: Shinji Furuzono, Tetsuya Kubota, Junki Taura.
✉email: karasawa.hiroshi.vr@daiichisankyo.co.jp; nkubota-tky@umin.ac.jp; kadowaki-3im@h.u-tokyo.ac.jp

Glucose transporter 4 (GLUT4), which is one of the glucose transporter isoforms that is expressed in the skeletal muscle, myocardium, and adipose tissue, is the rate-limiting transporter for glucose uptake and plays a crucial role in the maintenance of glucose homeostasis[1,2]. Subjects with type 2 diabetes show reduced glucose uptake by the skeletal muscle because of impaired GLUT4 translocation to the skeletal muscle cell surface[3]. It has been reported that GLUT4-overexpressing diabetic mice show markedly reduced plasma glucose levels under both fasting and postprandial conditions[4–6].

Although GLUT4 is stored in intracellular storage vesicles under basal conditions, insulin induces translocation of GLUT4 to the cell surface, facilitating glucose uptake[7,8]. Insulin activates Akt via insulin receptor substrate (IRS)s-phosphoinositide 3-kinase (PI3K)[9,10], and the activated Akt phosphorylates and consequently inhibits the proteins Akt substrate of 160 kDa (AS160) and TBC1 domain family member 1 (TBC1D1), both of which are Rab GTPase-activating proteins (GAPs); this results in activation of the Rab proteins and translocation of GLUT4 to the plasma membrane surface[11]. RAS-related C3 botulinum toxin substrate 1 (Rac1), another molecule downstream of PI3K, has been reported to promote GLUT4 translocation independently of the Akt-AS160/TBC1D1-Rab pathway. Rac1 stimulates reorganization of the cortical actin polymerization, which allows the GLUT4-containing vesicles to be inserted into the plasma membrane[12,13]. Insulin is known to regulate GLUT4 translocation via both the Akt-AS160-Rab pathway and Rac1-actin polymerization pathway[14,15]. In subjects with type 2 diabetes, both the insulin signaling pathways are impaired in the skeletal muscle, resulting in a reduction of insulin-induced glucose uptake by the skeletal muscle.

Contraction during exercise is another important enhancer of GLUT4 translocation in the skeletal muscle[16]. Upon increased glucose demand during exercise in the skeletal muscle, GLUT4 translocates to the cell surface to promote glucose supply to the skeletal muscle[17,18]. Exercise increases the AMP/ATP ratio caused by ATP consumption, leading to AMP-activated kinase (AMPK) activation. Despite the reported evidence of contraction inducing phosphorylation of TBC1D1 by activating AMPK[19] or of increased skeletal muscle glucose uptake by pharmacological activation of AMPK by AICAR[20], the significance of AMPK in exercise-stimulated glucose uptake in vivo remains controversial[21,22]. Recently, induction by Rac1 of NADPH oxidase 2-dependent production of reactive oxygen species was implicated in glucose uptake during exercise, through regulation of GLUT4 translocation[23,24]. Skeletal muscle contraction did not induce phosphorylation of IRS1 or PI3K[25]. Contraction-induced glucose uptake or GLUT4 translocation in the skeletal muscle was not inhibited by wortmannin, a PI3K inhibitor[26,27]. Moreover, combination of insulin and skeletal muscle contraction caused a further increase of GLUT4 translocation and glucose uptake as compared to insulin alone[27]. These data suggest that skeletal muscle contraction stimulates GLUT4 translocation independently of insulin.

In subjects with type 2 diabetes, skeletal muscle biopsy specimens obtained during a euglycemic insulin clamp showed impaired insulin signaling, observed as reduction in IRS1 phosphorylation and PI3K activity, in the skeletal muscle[28], while no effect was noted on the phosphorylation/activity of Akt[29]. Other studies have demonstrated reduced GLUT4 translocation and glucose uptake in subjects with type 2 diabetes[3,28]. Furthermore, it was reported that the reduced GLUT4 translocation in subjects with type 2 diabetes was improved by exercise[30,31]. These findings suggest that induction of GLUT4 translocation in the skeletal muscle could be a potential therapeutic target in patients with type 2 diabetes.

In the present study, we showed that the xanthene derivative DS20060511 induced skeletal muscle-specific GLUT4 translocation, independent of the actions of insulin. We used L6-myotubes expressing myc-tagged GLUT4 (L6-GLUT4myc) to screen our chemical compound library, and measured GLUT4 translocation to the cell surface by quantitative anti-myc immunoassay. The effects of the compound on the glucose uptake and whole-body glucose metabolism were examined in a series of in vitro and in vivo experiments. The mechanism of action of the compound was explored by investigating known signaling pathways involved in GLUT4 translocation induced by insulin and exercise. Finally, we evaluated the therapeutic potential of the compound in an obese and insulin-resistant mouse model of type 2 diabetes.

## Results

**The xanthene derivative, DS20060511, is a skeletal muscle cell-specific inducer of GLUT4 translocation.** We screened our chemical library, composed of more than 100,000 compounds, using L6-GLUT4myc myotubes, to identify compounds that would induce translocation of GLUT4 to the cell surface. Two completely different compounds were identified and both passed the counter assay to exclude compounds that would exert toxic effects, such as respiratory chain inhibition. Further in vitro assays revealed that one of the two compounds affected the Akt pathway, so that we finally selected the other, an original xanthene compound, as the hit compound with the potential effect of inducing GLUT4 translocation. Lead optimization of the hit compound finally yielded the more potent xanthene compound, DS20060511 (Fig. 1a and Supplementary Fig. 1). Treatment with DS20060511 increased GLUT4 translocation in differentiated L6-GLUT4myc myotubes in a concentration-dependent manner, as is the case with insulin treatment (Fig. 1b). However, while insulin treatment also increased GLUT4 translocation in differentiated 3T3-L1-GLUT4myc adipocytes, DS20060511 treatment had almost no effect on GLUT4 translocation in these adipocytes, suggesting that the induction of GLUT4 translocation by DS20060511 is specific to skeletal muscle cells (Fig. 1c). Consistent with these data, DS20060511 treatment significantly increased 2-DG uptake in a concentration-dependent manner in L6-GLUT4myc myotubes, as is the case with insulin treatment (Fig. 1d). Again, while insulin was shown to increase 2-DG uptake in differentiated 3T3-L1-GLUT4myc adipocytes, DS20060511 showed no such effect in the adipocytes (Fig. 1e). These data suggest that the xanthene compound DS20060511 promotes glucose uptake by skeletal muscle cell-specific activation of GLUT4 translocation.

**Treatment with DS20060511 decreases blood glucose levels by increasing skeletal muscle glucose uptake via inducting enhanced GLUT4 translocation in vivo.** To investigate the effects of DS20060511 on the glucose dynamics in vivo, DS20060511 was administered to normal mice. In mice that had continued access to food, oral administration of DS20060511 alone modestly, but statistically significantly, reduced the blood glucose levels, while in mice that had denied access to food overnight, the compound exerted no effect on the blood glucose levels (Fig. 2a, b). When it was administered prior to the oral glucose load in the oral glucose tolerance test (GTT), DS20060511 produced a dose-dependent suppression of rise of the blood glucose levels after an oral glucose load (Fig. 2c). Insulin secretion during oral GTT was rather significantly decreased in all the DS20060511-treated groups, suggesting that DS20060511 treatment decreases blood glucose levels independently of insulin secretion. DS20060511 treatment produced a significant increase in the uptake of [3H]-2-DG in the soleus and gastrocnemius

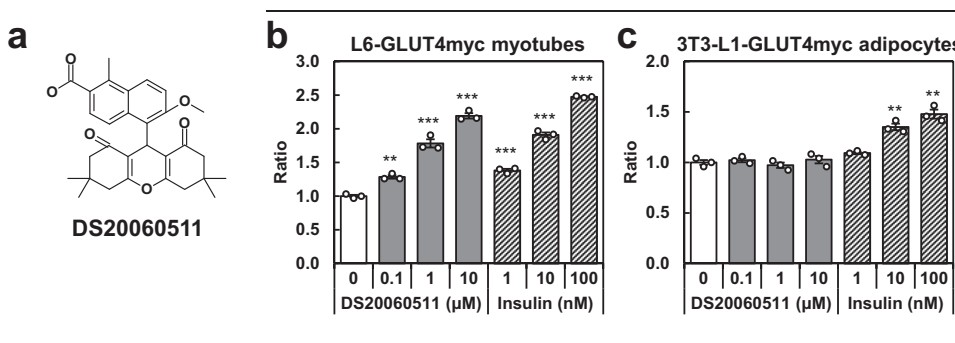

**Fig. 1 A xanthene derivative, DS20060511, induced skeletal muscle-specific GLUT4 translocation. a** Chemical structure of DS20060511. **b, c** Concentration-dependent induction of GLUT4 translocation by DS20060511 and insulin in L6-GLUT4myc myotubes (**b**) and 3T3-L1-GLUT4myc adipocytes (**c**). **d, e** 2-DG uptake evaluated in L6-GLUT4myc myotubes (**d**) and 3T3-L1-GLUT4myc adipocytes (**e**). Values shown are means ± SEM, $n = 3$. **$P < 0.01$, ***$P < 0.001$ vs. control by one-way ANOVA followed by Dunnett's test.

muscles, but not in heart or white adipose tissue (WAT) during intraperitoneal GTT (Fig. 2d). Western blot analysis revealed increased GLUT4 protein expression levels in the plasma membrane fraction of the skeletal muscles in the DS2006511-treated group as seen in an insulin-treated group (Fig. 2e). These data suggest that DS20060511 treatment decreases the blood glucose levels by increasing skeletal muscle glucose uptake via inducing GLUT4 translocation in vivo.

**Pharmacokinetic evaluation of DS20060511 in mice.** Changes in the plasma concentration and distribution of DS20060511 to possible target organs/tissues were examined in normal mice. The levels of systemic exposure to DS20060511 after oral administration of the compound was dose dependent, and the maximal concentrations at 30 min after administration of 1, 10, and 30 mg kg$^{-1}$ were 0.6, 16.5, and 71.4 μM, respectively (Supplementary Fig. 2a). Measurement of the DS20060511 concentrations in tissues at 75 min after oral administration (30 mg kg$^{-1}$) revealed almost comparable concentrations among the skeletal muscle, WAT, and heart (Supplementary Fig. 2b). Consistent with its stable pharmacokinetic profile, the metabolic stability of the compound in the liver microsomal fraction was high (89% and 79% compound remaining after 1 h incubation with the mouse and human liver microsomal fraction, respectively).

**The glucose-lowering effect of DS20060511 is dependent on GLUT4.** To confirm that the glucose-lowering effect of DS20060511 is mediated by GLUT4, we administered DS20060511 to GLUT4KO mice. GLUT4 protein expression was undetectable in the skeletal muscle, heart, and WAT of the GLUT4KO mice (Supplementary Fig. 3). While DS20060511 treatment caused a significant decrease of the blood glucose and plasma insulin levels in the wild-type (WT) mice during oral

GTT, these effects were completely abolished in the GLUT4KO mice (Fig. 3a). DS20060511 treatment significantly increased the 2-DG uptake by isolated soleus and extensor digitorum longus (EDL) muscles of the WT mice, whereas no such increase in muscle uptake was observed in the isolated muscles of the GLUT4KO mice treated with DS20060511 (Fig. 3b). These data confirm that the glucose-lowering effect of DS20060511 is mediated by GLUT4 in the skeletal muscle.

**Treatment with DS20060511 induces GLUT4 translocation without activation of the IR-IRS1-PI3K-Akt-AS160 and -PI3K-Rac1 pathways.** The insulin-induced GLUT4 translocation are activated by (1) the IR-IRS1-PI3K-Akt-AS160 pathway[32], and (2) the IR-IRS1-PI3K-Rac1 pathway[15] in the skeletal muscle. Insulin binds the IR, which results in the activation of IRS1, PI3K, and Akt. Activated Akt inhibits the Rab GTPase-activating protein (GAP) AS160, which results in activation of the Rab proteins and translocation of GLUT4 to the plasma membrane[33]. On the other hand, Rac1 is activated by PI3K and promotes actin remodeling, resulting in translocation of GLUT4[12]. We examined whether DS20060511 treatment increases GLUT4 translocation in the skeletal muscle via these pathways. Although the IRβ-subunit and IRS1 were phosphorylated in the skeletal muscles of the insulin-treated mice, no such phosphorylation of these proteins was observed after DS20060511 treatment (Fig. 4a). Similarly, while insulin treatment induced phosphorylation of Akt and AS160, DS20060511 treatment had no such effect (Fig. 4b). We next performed immunofluorescence microscopy to investigate whether DS20060511 might promote actin polymerization. Although strong staining of GLUT4 at the cell surface was observed following both insulin and DS20060511 treatment, actin polymerization was observed only following insulin treatment in the differentiated L6-GLUT4myc myotubes (Fig. 4c). Moreover,

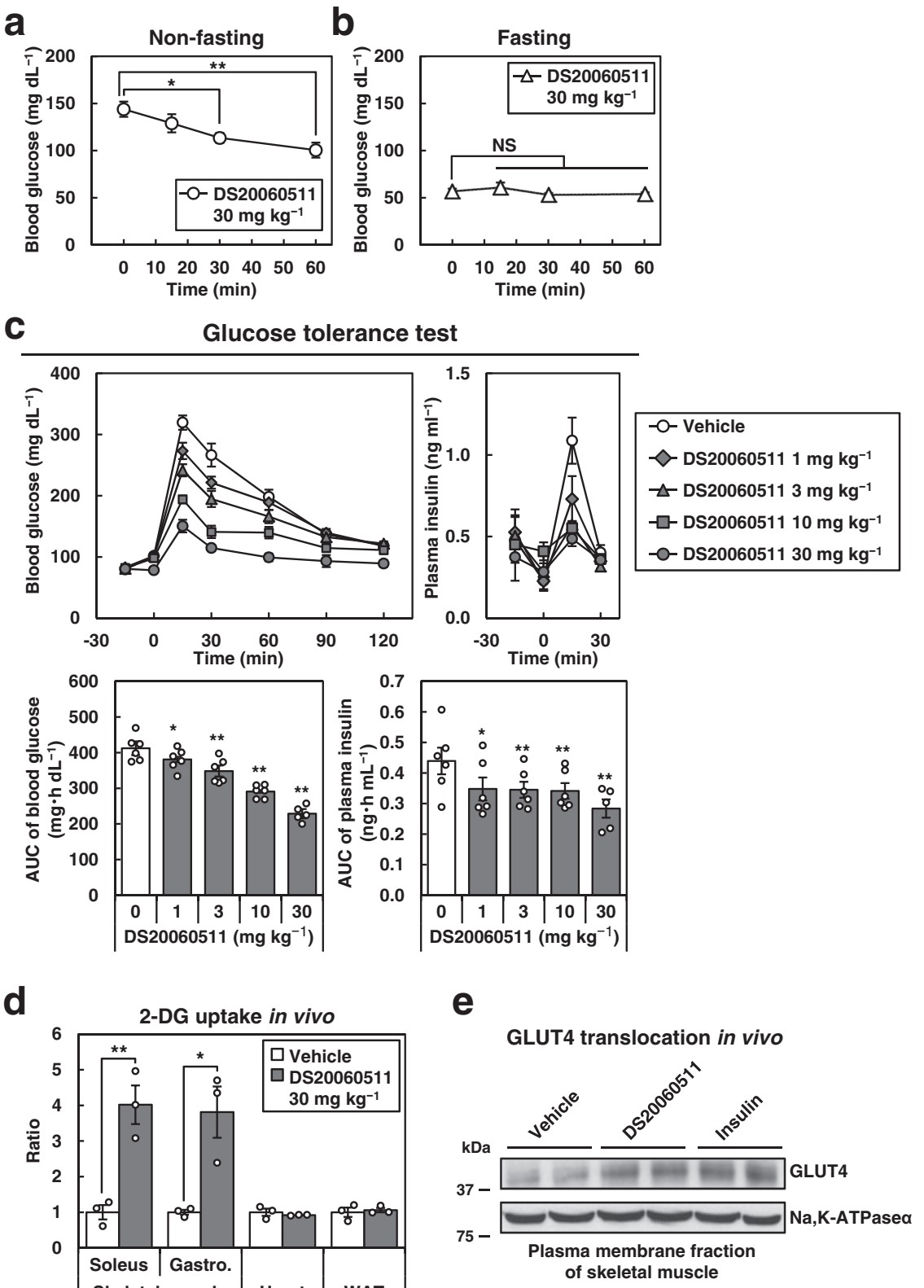

although GLUT4 translocation was induced by both insulin and DS20060511 treatment, Latrunculin B, an actin polymerization inhibitor, only suppressed GLUT4 translocation induced by insulin, but not that induced by DS20060511 treatment (Fig. 4d). Co-treatment of DS20060511 and insulin resulted in an additive increase of GLUT4 translocation in the

L6-GLUT4myc myotubes, even at the insulin concentration at which GLUT4 translocation by insulin alone was saturated (Fig. 4e). Consistent with these data, 2-DG uptake induced by insulin was also additively increased by concomitant treatment with DS20060511 in isolated skeletal muscles (Fig. 4f). In fact, blood glucose levels were reduced to a greater degree after

**Fig. 2 Treatment with DS20060511 decreases the blood glucose levels via inducing GLUT4 translocation and increasing skeletal muscle glucose uptake. a, b** Blood glucose levels after treatment with DS20060511 (30 mg kg$^{-1}$) in C57BL/6 mice that had received continued access to food (**a**) and mice that had been denied access to food overnight (**b**) ($n = 8$). Values shown are means ± SEM. **$P < 0.01$ vs. 0 min by one-way ANOVA followed by Dunnett's test. **c** Blood glucose and plasma insulin levels during oral GTT in the C57BL/6 mice ($n = 5$–6). The mice received oral administration of vehicle or DS20060511 at the indicated dose, 15 min prior to glucose administration (1.5 g kg$^{-1}$). Values shown are means ± SEM. *$P < 0.05$, **$P < 0.01$ vs. vehicle by one-way ANOVA followed by Williams' test. **d** [$^3$H]-2-DG uptake in the soleus muscle, gastrocnemius muscle (Gastro.), heart, and white adipose tissue (WAT) at 60 min during the intraperitoneal GTT in the C57BL/6 mice ($n = 3$). The mice received oral administration of vehicle or DS20060511 (30 mg kg$^{-1}$), 15 min prior to glucose administration (1 g kg$^{-1}$ glucose containing [$^3$H]-2-DG). Values shown are means ± SEM. *$P < 0.05$, **$P < 0.01$ vs. vehicle by the $t$-test. **e** Protein levels of GLUT4 and Na,K-ATPaseα in the plasma membrane fraction of the triceps surae muscle excised from the C57BL/6 mice ($n = 2$) treated with DS20060511 (10 mg kg$^{-1}$), insulin (5 U kg$^{-1}$), or saline as vehicle, via the inferior vena cava 10 min after the treatment. Uncropped blots for **e** can be found in Supplementary Fig. 6.

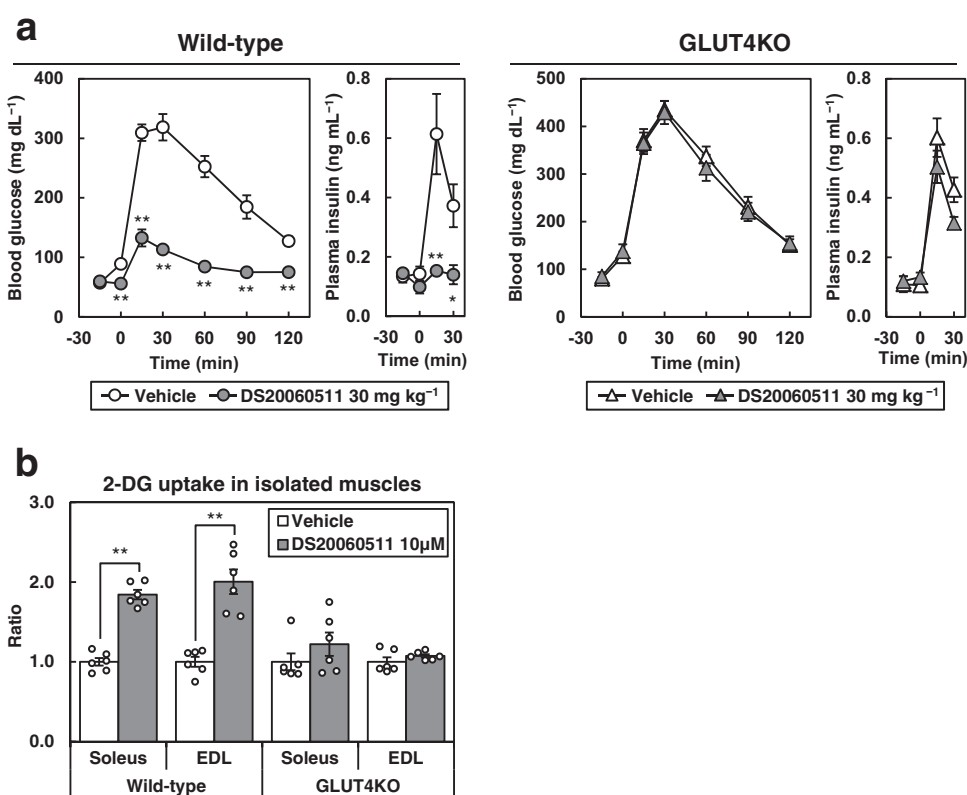

**Fig. 3 The glucose-lowering effect of DS20060511 is totally dependent on GLUT4. a** Blood glucose and plasma insulin levels during oral GTT in wild-type (WT, $n = 5$) and GLUT4 knockout (KO, $n = 6$) mice. Mice received oral administration of vehicle or DS20060511 (30 mg kg$^{-1}$), 15 min prior to the glucose administration (1.5 g kg$^{-1}$). Values shown are means ± SEM. *$P < 0.05$, **$P < 0.01$ vs. vehicle by the $t$-test. **b** DS20060511-stimulated [$^3$H]-2-DG uptake in the isolated soleus and EDL muscles excised from WT ($n = 6$) and KO ($n = 6$) mice. Values shown are means ± SEM. **$P < 0.01$ vs. vehicle by the $t$-test.

combined DS20060511 plus insulin treatment as compared to that after insulin treatment alone in streptozotocin (STZ)-treated mice (Fig. 4g). These data suggest that activation of neither the IR-IRS1-PI3K-Akt-AS160 pathway nor the IR-IRS1-PI3K-Rac1 pathway is involved in the GLUT 4 translocation induced by DS20060511 treatment.

**Treatment with DS20060511 increases glucose oxidation during exercise.** Since exercise, like insulin, is well known to enhance GLUT4 translocation and increase glucose uptake into the skeletal muscle[34], we next investigated the effect of DS20060511 treatment on the exercise endurance capacity and fuel oxidation during exercise by calorimetry. During the stepwise treadmill exercise, the VO$_2$ gradually increased in both the vehicle- and DS20060511-treated groups (Supplementary Fig. 4a), and the exercise endurance capacity was also comparable between the two groups (Supplementary Fig. 4b). After a while from the beginning of running, the DS20060511-treated group started to show

relatively higher respiratory exchange ratio (RER) to the vehicle-treated group (Fig. 5a); furthermore, the estimated glucose oxidation during the test was significantly higher in the DS20060511-treated mice as compared to the vehicle-treated mice, while the fat oxidation was significantly lower (Fig. 5b, c). Thus, DS20060511 increased glucose oxidation during exercise. The blood glucose levels decreased significantly after exercise in the DS20060511-treated mice, but did not dip to the hypoglycemia range. The blood lactate levels were comparable between the two groups (Supplementary Fig. 4c).

**Treatment with DS20060511 has no effects on AMPK phosphorylation.** Based on the finding that DS20060511 increased glucose utilization in the skeletal muscle during exercise, its effects combined with those of muscle contraction were further evaluated using isolated skeletal muscle specimens. 2-DG uptake was elevated to a greater degree following electrical muscle stimulation combined with DS20060511 treatment as compared

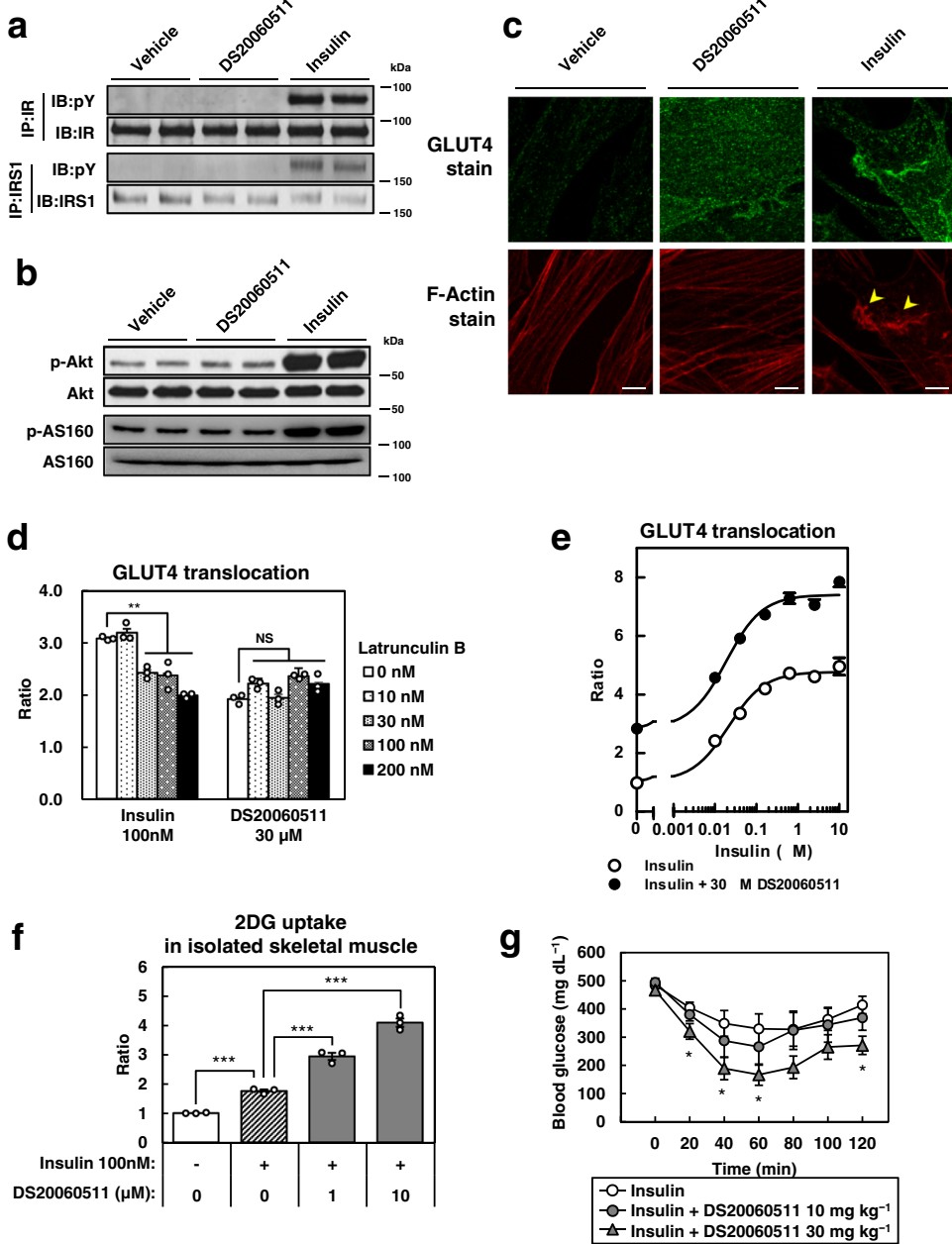

**Fig. 4 The mechanism underlying DS20060511-induced GLUT4 translocation is distinct from that of insulin-induced GLUT4 translocation. a**, **b** Phosphorylation of IRβ, IRS1, Akt (Ser473), and AS160 of the triceps surae muscle excised from C57BL/6 mice ($n = 2$) treated with DS20060511 (10 mg kg$^{-1}$), insulin (5 U kg$^{-1}$), or saline as vehicle, via inferior vena cava 10 min after the treatment. **c** Fluorescence immunostaining of cell surface GLUT4 and intracellular actin fibers in L6-GLUT4myc myotubes treated with 30 μM of DS20060511 or 100 nM of insulin. Arrowheads indicate the characteristic ruffled structure of the polymerized actin and actin-associated surface GLUT4. **d** GLUT4 translocation activity of 30 μM DS20060511 or 100 nM insulin in the presence of the actin polymerization inhibitor, Latrunculin B, at the indicated concentrations. Values shown are means ± SEM, n = 3. **$P < 0.01$ vs. 0 nM Latrunculin B by one-way ANOVA followed by Dunnett's test. **e** Concentration-dependent insulin-stimulated GLUT4 translocation in L6-GLUT4myc myotubes with or without 30 μM DS20060511 ($n = 3$). **f** Concentration-dependent DS20060511-stimulated 2-DG uptake with 100 nM insulin in isolated muscles from C57BL/6 mice ($n = 3$). Values shown are means ± SEM. ***$P < 0.001$ by one-way ANOVA followed by Tukey's test. **g** Blood glucose levels during ITT in STZ-treated C57BL/6 mice ($n = 6$–7). Vehicle or indicated dose of DS20060511 was given orally at the same time as 0.1 U kg$^{-1}$ insulin injection intraperitoneally. Values shown are means ± SEM. *$P < 0.05$ vs. vehicle by one-way ANOVA followed by Dunnett's test. **c** Scale bar in all panels, 5 μm. Uncropped blots for **a** and **b** can be found in Supplementary Fig. 6.

that following electrical muscle stimulation alone without DS20060511 treatment (Fig. 6a). Although recent findings suggest that AMPK plays no role in the GLUT4 translocation and glucose uptake in the muscle observed during exercise[16,22], activation of AMPK by electrical stimulation[21], as well as by AICAR[20], could increase the glucose uptake in isolated skeletal muscle. We examined the phosphorylation of AMPK following DS20060511

treatment by western blotting in isolated skeletal muscle. Although the AMPK phosphorylation level was elevated by electrical muscle stimulation, no such change was observed after DS20060511 treatment (Fig. 6b). The AMPK phosphorylation level in the skeletal muscle remained unchanged after DS20060511 treatment as compared to that before treatment in vivo, even under the no-exercise condition (Fig. 6c). These data

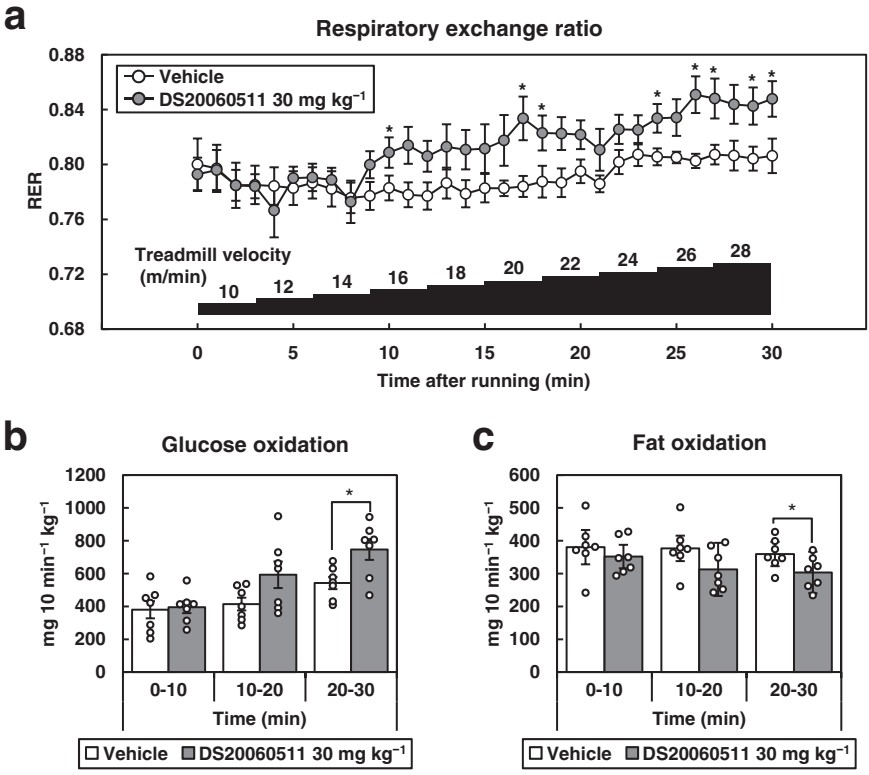

**Fig. 5 DS20060511 increases glucose oxidation during exercise. a–c** Respiratory exchange ratio (RER), estimated glucose oxidation, and fat oxidation during stepwise treadmill running in C57BL/6 mice (n = 7). Vehicle or DS20060511 (30 mg kg$^{-1}$) was given orally 15 min before starting running. Treadmill started from the velocity of 10 m min$^{-1}$ and increased by 2 m min$^{-1}$ each 3 min. Values shown are means ± SEM. *$P < 0.05$ vs. vehicle by the *t*-test.

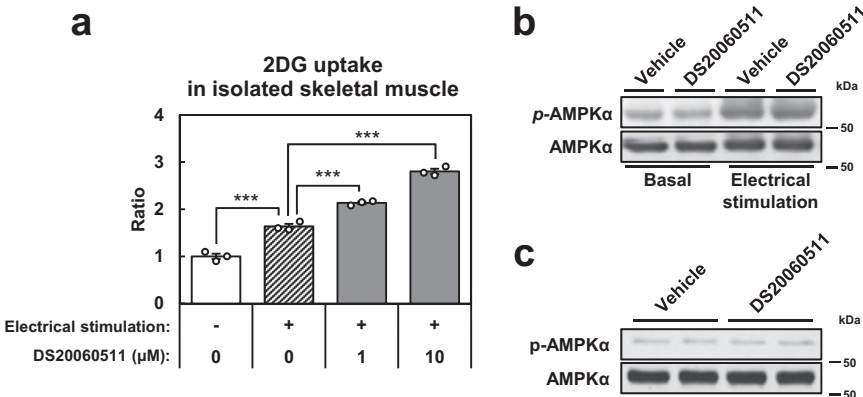

**Fig. 6 DS20060511 increases glucose uptake without affecting AMPK phosphorylation. a** Concentration-dependent DS20060511-stimulated 2-DG uptake with muscle contraction (5 Hz electrical stimulation) in isolated muscles from C57BL/6 mice (n = 3). ***$P < 0.001$ by one-way ANOVA followed by Tukey's test. **b** Muscle contraction (5 Hz electrical stimulation) induced AMPK (Thr172) phosphorylation with or without 10 µM DS20060511 in isolated muscles from C57BL/6 mice. **c** Phosphorylation levels of AMPKα of the Triceps surae muscles excised from C57BL/6 mice (n = 2) treated with DS20060511 (10 mg kg$^{-1}$) or saline as vehicle via inferior vena cava 10 min after the treatment. Uncropped blots for **b** and **c** can be found in Supplementary Fig. 7.

suggest that the increase in glucose uptake induced by DS20060511 is independent of AMPK activation.

**Treatment with DS20060511 decreases the blood glucose in an eNOS-independent manner**. It has been shown that sodium nitroprusside (SNP), a nitric oxide (NO) donor, increases glucose uptake in the skeletal muscle and that this increase is not inhibited by the PI3K inhibitor, wortmannin[35]. In addition, exercise-induced glucose uptake by the skeletal muscle was not suppressed by the NO inhibitor N$^G$-monomethyl-L-arginine (L-NMMA)[35]. These data suggest that NO induces glucose uptake by the skeletal muscle via a mechanism that is distinct from both the insulin and exercise signaling pathways. Endothelial nitric oxide synthase, which is a major enzyme generating NO, is expressed in the skeletal muscle. Glucose uptake has been reported to be impaired in isolated skeletal muscles from eNOSKO mice[36]. To investigate the mechanism underlying the

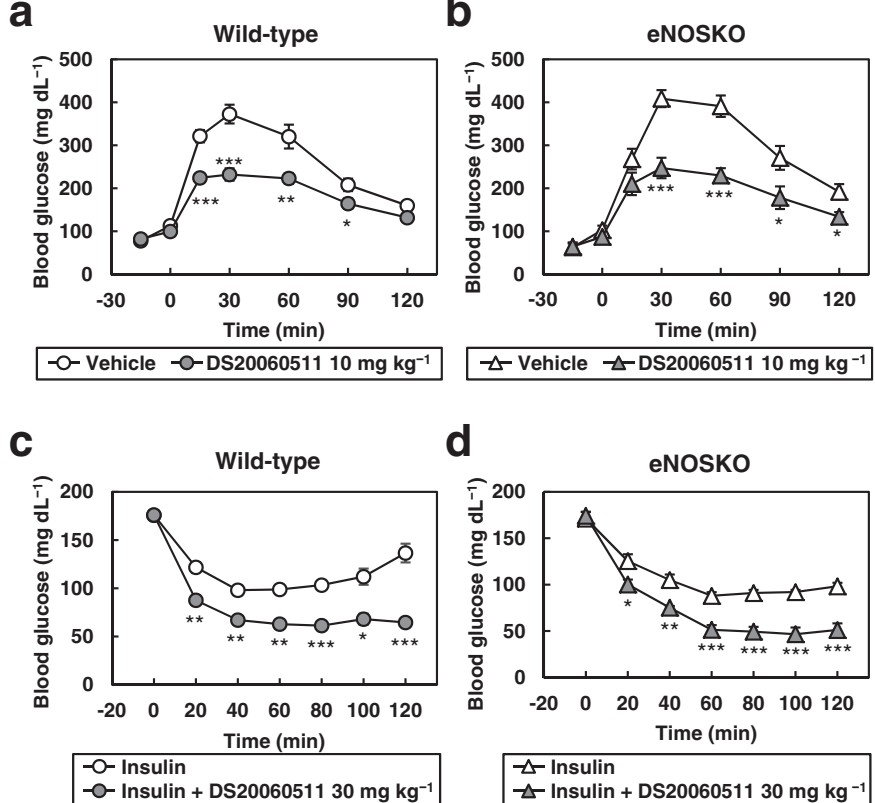

**Fig. 7 DS20060511 treatment decreases blood glucose levels in both WT and eNOSKO mice during oral GTT and ITT. a, b** Blood glucose levels during oral GTT in wild-type (WT, $n = 5$) and eNOS-knockout (KO, $n = 5–6$) mice. Mice received vehicle or DS20060511 (10 mg kg$^{-1}$) orally 15 min before the glucose administration (3.0 g kg$^{-1}$). **c, d** Blood glucose levels during ITT in WT ($n = 4$) and KO ($n = 5$) mice. Vehicle or DS20060511 (30 mg kg$^{-1}$) was given orally at the same time as 0.5 U kg$^{-1}$ insulin injection intraperitoneally. Values shown are means ± SEM. *$P < 0.05$, **$P < 0.01$, ***$P < 0.001$ vs. vehicle by the $t$-test.

increase in glucose uptake by the skeletal muscle induced by DS20060511, we administrated DS20060511 to eNOSKO mice. Treatment with DS20060511 significantly decreased blood glucose levels in both the WT and eNOSKO mice during oral GTT (Fig. 7a, b). Although the blood glucose levels were reduced by insulin treatment, the blood glucose levels were reduced even further after DS20060511 treatment, in both the WT and eNOSKO mice, similarly (Fig. 7c, d). These data suggest that the glucose-lowering effect of DS20060511 is exerted in an eNOS-independent manner.

**Acute and chronic DS20060511 treatment improves glucose intolerance in obese diabetic mice.** To investigate whether DS20060511 treatment can attenuate glucose intolerance in mice with diet-induced obesity and insulin resistance, we conducted oral GTT in high-fat diet (HFD)-fed mice after DS20060511 treatment. Treatment with DS20060511 significantly decreased the blood glucose levels in the HFD-fed mice to the same levels as those observed in normal-chow diet-fed mice during oral GTT (Fig. 8a). The plasma insulin levels were rather decreased in the DS20060511-treated HFD-fed mice (Fig. 8a). Suppression of insulin-induced 2-DG uptake in isolated skeletal muscle from HFD-fed mice was completely restored by DS20060511 treatment (Fig. 8b). These data suggest that acute DS20060511 treatment improves glucose intolerance in mice with diet-induced obesity and insulin resistance. Next, we investigated the effect of chronic DS20060511 treatment in genetically obese diabetic (*db/db*) mice. The blood glucose levels decreased significantly from the first to the 28th day of DS20060511 treatment in the *db/db* mice

(Fig. 8c, d). Consistent with these data, the hemoglobin A1c (HbA1c) value was also significantly reduced after chronic DS20060511 treatment (Fig. 8e). There were no statistically significant differences in the body weight, food intake, fasting blood glucose level, or fasting plasma insulin levels between the DS20060511- and vehicle-treated *db/db* mice (Supplementary Fig. 5a). There were also no significant changes in the tissue weights of the muscle, heart, WAT, and liver, or in the glycogen contents of the muscle, heart, and liver (Supplementary Fig. 5b, c). These data suggest that both acute and chronic DS20060511 treatment improves diabetes via restoring impaired skeletal muscle glucose uptake.

## Discussion

We explored our chemical libraries using L6-GLUT4myc myotubes for a new drug to treat type 2 diabetes, and discovered the xanthene compound, DS20060511. DS20060511 increased GLUT4 translocation in differentiated L6-GLUT4myc myotubes, but not in differentiated 3T3-L1-GLUT4myc adipocytes, suggesting that it acts primarily in the skeletal muscles. Consistently, in vivo, DS20060511 induced 2-DG uptake in the soleus and gastrocnemius muscles, but not in the heart or adipose tissue. Insulin promotes glucose uptake in the adipose tissue as well as skeletal muscle, which inevitably leads to obesity. However, DS20060511 enhances glucose uptake only in the skeletal muscle, and reduces insulin secretion by suppressing the rise in blood glucose levels after glucose loading, thereby suppressing the development of obesity; thus, the compound appears to also offer promise as a drug for the prevention of obesity. DS20060511

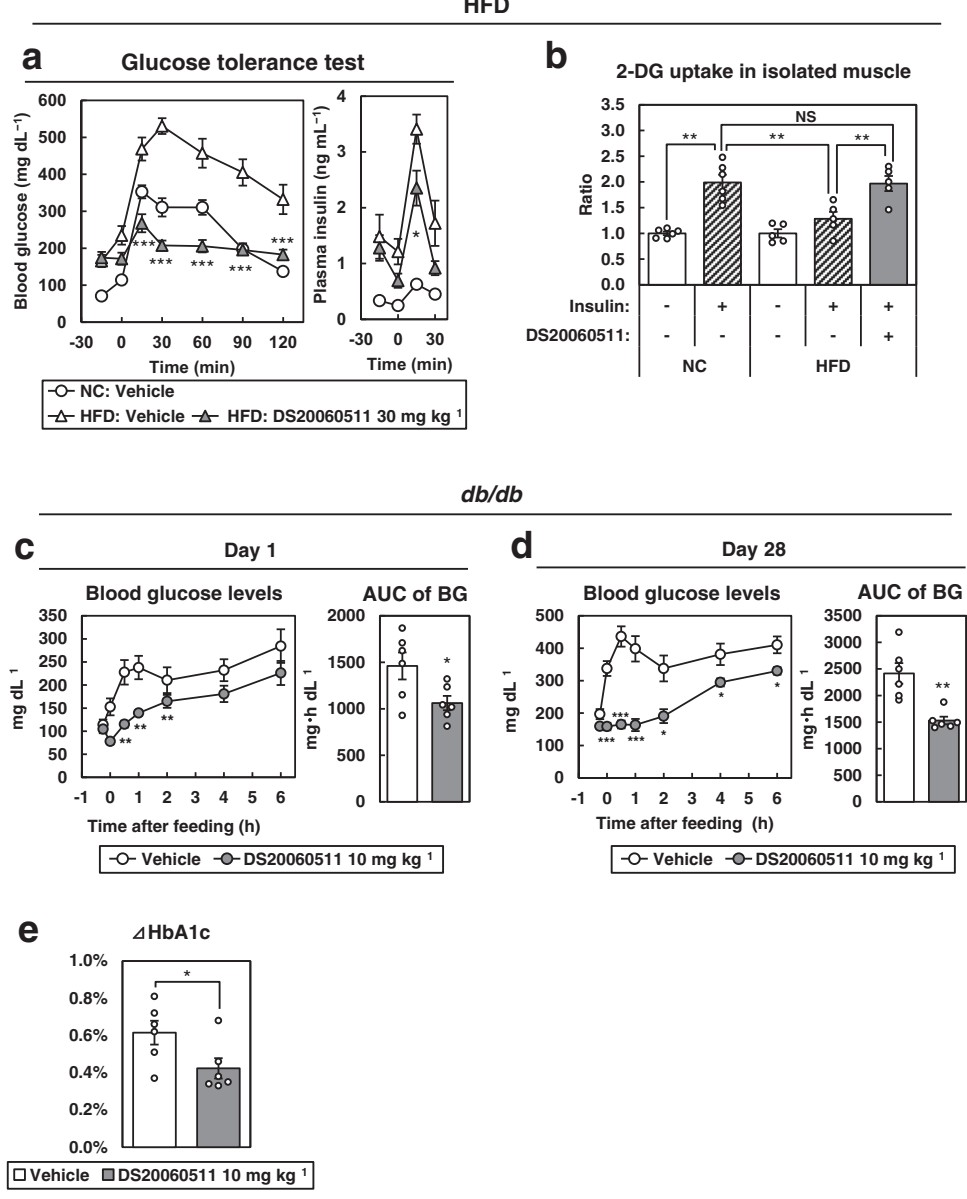

**Fig. 8 Acute and chronic DS20060511 treatment improves glucose intolerance in HFD-fed and *db/db* mice. a** Blood glucose and plasma insulin levels during an oral GTT in normal-chow (NC)- and high-fat diet (HFD)-fed mice ($n = 5$). Vehicle or DS20060511 (30 mg kg$^{-1}$) was given orally 15 min before the oral glucose administration (1.5 g kg$^{-1}$). Values shown are means ± SEM. *$P < 0.05$, **$P < 0.01$, ***$P < 0.001$ vs. the HFD vehicle by the *t*-test. **b** Effects of 10 μM DS20060511 and 100 nM insulin on 2-DG uptake in muscles isolated from NC-fed ($n = 6$) and HFD-fed ($n = 5$) mice. Values shown are means ± SEM. **$P < 0.01$ by one-way ANOVA followed by Tukey's test. **c**, **d** Changes in the blood glucose levels and AUC on day 1 and day 28 during refeeding ($n = 6$) in *db/db* mice treated chronically with DS20060511 (10 mg kg$^{-1}$ day$^{-1}$). Values shown are means ± SEM. *$P < 0.05$, **$P < 0.01$, ***$P < 0.001$ vs. vehicle by the *t*-test. **e** Change of the HbA1c levels in the *db/db* ($n = 6$) mice at 4 weeks. Values shown are means ± SEM. *$P < 0.05$ vs. vehicle by the *t*-test.

reduced the blood glucose levels in obese diabetic mice, without causing hyperphagia, body weight gain, or hypoglycemia, and without increasing insulin secretion. In addition, DS20060511 does not appear to lower the blood glucose level in the fasting state, indicating the relatively low risk of hypoglycemia associated with the use of this compound. These characteristics could be preferable to a safe and effective drug for the treatment of type 2 diabetes.

The glucose-lowering effect of DS20060511 was completely abolished in GLUT4KO mice, indicating that DS20060511 increases glucose uptake in a GLUT4-dependent manner. Interestingly, DS20060511 failed to activate upstream insulin signaling including phosphorylation of AS160 and actin remodeling or the AMPK pathway, which are also known to increase GLUT4

translocation in the skeletal muscle. Moreover, when administered in combination with insulin, DS20060511 further enhanced glucose uptake into the skeletal muscle in both normal and insulin-resistant mice, and further reduced the blood glucose levels in a mouse model of STZ-induced type 1 diabetes. DS20060511 also enhanced whole-body glucose oxidation during exercise, associated with increased glucose uptake and utilization in the skeletal muscle[16]. Thus, DS20060511 may act as an antidiabetic agent with an entirely novel mechanism of action in patients with impaired actions of insulin in the skeletal muscle and those with either type 1 or 2 diabetes receiving insulin and/or exercise therapy.

Some compounds have also been previously reported to induce GLUT4 translocation. Novel pyridazine compounds

were reported to strongly induce GLUT4 translocation in L6-myotubes and to show a significant blood glucose-lowering effect in a mouse model of severe diabetes[37]. Proton uncouplers, such as 2,4-dinitrophenol, are well known to induce GLUT4 translocation in accordance with a rapid drop in the intracellular ATP levels[38]. However, unlike DS20060511, these compounds promote GLUT4 translocation via the PI3K or AMPK pathway. The NO-donating small molecule NCX 4016 has been reported to induce GLUT4 translocation in 3T3-L1 adipocytes, but not in skeletal muscle cells[39]. These findings suggest that a skeletal muscle-specific GLUT4 translocation enhancer like DS20060511 has never previously been reported.

Why does DS2006051 act selectively on the skeletal muscle? The target molecule of DS2006051 may be selectively expressed in the skeletal muscle. The amount of GLUT4 on the cell surface is determined by the balance between exocytosis from the intracellular storage vesicles and endocytosis from the cell membrane. DS2006051 may promote exocytosis or suppress endocytosis of GLUT4 via target molecule activation. To investigate the selective target of DS20060511 in the skeletal muscle and L6-myotubes, we adopted three different approaches: radiolabeled compound binding, compound-immobilized beads purification, and UV photo-crosslinking of a compound to the target. Radiolabeled or chemically modified compounds were allowed to react with samples prepared from skeletal muscle tissue or L6-GLUT4myc myotubes, such as lysates, microsomes, or living cells. After enrichment and purification matched for each approach, the samples were analyzed by LC-MS/MS. Unfortunately, we could identify no specific target molecule that bound to DS20060511 with a high affinity. Further investigation to identify the molecular target of DS20060511 and also the signaling pathway involved, such as Rac1 or NADPH oxidase 2-associated reactive oxygen species production, is needed.

In conclusion, we identified a novel xanthene compound, DS20060511, and demonstrated that treatment with DS20060511 induced GLUT4 translocation independently of canonical insulin signaling and AMPK activity, to enhance glucose uptake by the skeletal muscle. Moreover, DS20060511 treatment also ameliorated glucose intolerance in obese diabetic mice. Although we could not identify the specific target molecule of DS20060511 on the skeletal muscle cell, further studies with the compound would help to develop a novel drug for type 2 diabetes.

## Methods

**Chemicals**. DS20060511 (Fig. 1a, molecular formula $C_{30}H_{32}O_6 \cdot C_4H_{11}N$, molecular weight 561.71) was obtained from the Medicinal Chemistry Research Laboratories, Daiichi Sankyo Co., Ltd., Japan. The concentrations and doses of DS20060511 used for the in vitro and in vivo experiments in this study were selected based on the results of preliminary concentration- and dose-finding experiments, including the GLUT4 translocation assay in L6-GLUT4myc myotubes and GTT, respectively. 2-Deoxy-D-[l,2-$^3$H]-glucose ([$^3$H]-2-DG, ART0103A) and D-[1-$^{12}$C]-mannitol ([$^{12}$C]-mannitol, ARC0127A) were purchased from American Radiolabeled Chemicals, St. Louis, MO. Streptozotocin was purchased from FUJIFILM Wako Pure Chemical, Osaka, Japan. Insulin (Humulin) used in the in vivo experiments was purchased from Eli Lilly, Indianapolis, IN. Insulin solution human (I9278) used in the in vitro experiments was purchased from Sigma–Aldrich, St. Louis, MO.

**Animals**. C57BL/6 and *db/db* mice were purchased from CLEA Japan, Tokyo, Japan. GLUT4- knockout mice were generated at RIKEN BioResource Center, Tsukuba, Japan, according to a previously described method[40]. eNOS-knockout mice were kindly provided by Dr. Masato Tsutsui[41,42]. The mice were group-housed under controlled illumination (12:12 h light-dark cycle) and temperature (23 ± 2 °C) conditions and given free access to normal chow and water, unless otherwise specified. The HFD-fed obese mice were prepared by feeding C57BL/6 mice a HFD for at least 8 weeks from 6 weeks of age. The CE2 normal rodent chow (CLEA Japan) had a calorie ratio of protein: fat: carbohydrate of 29.5: 11.9: 58.5, with a metabolic calorie content of 3.4 kcal g$^{-1}$. The HFD32 (CLEA Japan) used as the HFD had a calorie ratio of protein: fat: carbohydrate of 20.1: 56.7: 23.2, with a metabolic calorie content of 5.1 kcal g$^{-1}$. The sources of the fat were safflower oil

and beef tallow (20.0% and 15.9% in weight, respectively). Male mice were used for all the experiments in this study. The animal care and experimental procedures used in the study were approved by The University of Tokyo Animal Care Committee (Approval number: 27-3), and the study was carried out in accordance with the Animal Experimentation Guidelines of Daiichi Sankyo Co., Ltd (Approved number: 20000411).

**Streptozotocin (STZ)-treated C57BL/6 mouse model**. C57BL/6 mice deprived of access to food overnight received an intraperitoneal injection of STZ (125 mg kg$^{-1}$) and a repeat injection a week later. Thereafter, their blood glucose levels were monitored and insufficient mice (blood glucose levels of under 400 mg dL$^{-1}$ or blood glucose levels after overnight food deprivation of under 200 mg dL$^{-1}$) were excluded from the experiment. Blood glucose levels were measured using Glutest Every (Sanwa Kagaku Kenkyusho, Nagoya, Japan) in blood samples collected from the tail vein.

**Cell culture and differentiation**. The L6-GLUT4myc rat myoblast cell line was obtained from Dr. Amira Klip and Dr. Philip Bilan, through a license granted by The Hospital for Sick Children, 555 University Avenue, Toronto, Ontario, Canada M5G 1 × 8; or purchased from Kerafast, Boston, MA (ESK202)[43]. The 3T3-L1-GLUT4myc fibroblast cell line was provided by Dr. Tomoyuki Yuasa, Tokushima University[44]. L6-GLUT4myc myoblasts were grown in MEMα (32571036, Thermo Fisher Scientific, Waltham, MA) supplemented with 10% FBS and 1% antibiotics, and then induced to differentiate into myotubes in MEMα supplemented with 2% FBS and 1% antibiotics for 5–8 days. 3T3-L1-GLUT4myc fibroblasts were induced to differentiate into adipocytes as described previously,[45] with minor modifications. Briefly, cells were grown to confluence in growth medium: DMEM (10569010, Thermo Fisher Scientific) supplemented with 10% FBS and 1% antibiotics, and then induced to differentiate into adipocytes in growth medium supplemented with 1 μM dexamethasone, 2 μM rosiglitazone, 0.5 mM isobutylmethylxanthine, and 10 μg mL$^{-1}$ insulin for 2 days. The adipocytes were then cultured in growth medium supplemented with 10 μg mL$^{-1}$ insulin for a few days before being used for the experiments.

**Detection and quantitation of cell surface GLUT4 using anti-Myc antibody**. Cell surface GLUT4 levels in the L6-GLUT4myc myoblasts and myotubes were determined by antibody binding assay as described previously[43,44], with minor modifications. Briefly, cells in a 96- or 24-well plate were starved in serum-free MEMα (0.1% BSA and 1% antibiotics) for 3 h and then treated with the indicated concentrations of DS20060511 and/or insulin for 30 min at 37 °C. After fixation with 4% paraformaldehyde, the cells were incubated with 1% glycine in PBS at 4 °C for 10 min, and blocked with 10% normal goat serum and 3% BSA in PBS at 4 °C for 30 min. The cells were then incubated with anti-c-Myc antibody (sc-40, 1:500) diluted with blocking buffer at 4 °C for 1 h, washed with cold PBS(+), incubated with HRP-conjugated anti-mouse IgG diluted with blocking buffer at 4 °C for 30 min, and then washed again. SuperSignal ELISA Pico Chemiluminescent Substrate (Thermo Fisher Scientific) was added and the luminescent signal was measured. To investigate GLUT4 translocation in the 3T3-L1-GLUT4myc adipocytes, the cells were prepared in a 24-well plate, and anti-c-Myc antibody (sc-789, 1:1000), HRP-conjugated anti-rabbit IgG and SIGMAFAST OPD regent (Sigma–Aldrich) were added for optical detection of the cell surface GLUT4myc levels. The GLUT4 translocation activity was normalized to that in the vehicle-treated group.

**In vitro cellular 2-DG uptake**. L6-GLUT4myc myotubes and 3T3-L1-GLUT4myc adipocytes were incubated in the wells of a 24-well plate containing serum-free medium for 3 h at 37 °C and then incubated in glucose-free medium for 30 min at 37 °C. The cells were treated with the indicated concentrations of DS20060511 or insulin for 30 min at 37 °C, followed by the addition of 1 mM 2-deoxy-D-glucose (2-DG) and 0.3 μCi mL$^{-1}$ [$^3$H]-2-DG for 10 min. 2-DG uptake was measured with a liquid scintillation counter (Tri-Carb 2810TR, PerkinElmer, Waltham, MA) and normalized to the level in the vehicle-treated group.

**Glucose tolerance test (GTT)**. Mice that had been denied access to food overnight received the indicated oral dose of DS20060511 or vehicle 15 min prior to the glucose load (1.5 g kg$^{-1}$, except for eNOS-knockout mice, which received 3.0 g kg$^{-1}$). Then, after the oral glucose load, the blood glucose levels were measured using Glutest Every in samples obtained from the tail vein at the indicated timepoints. Plasma insulin concentrations were also measured with an ELISA kit (Morinaga Institute of Biological Science, Yokohama, Japan) in blood samples collected from the tail vein at the indicated timepoints.

**Insulin tolerance test (ITT)**. Mice received the indicated oral dose of DS20060511 or vehicle at the same time as the intraperitoneal injection of insulin (Humulin) at the indicated dose. Blood glucose levels were measured using Glutest Every in blood samples collected from the tail vein at the indicated timepoints.

**In vivo 2-DG uptake**. Tissue glucose uptake was examined by measuring the uptake of [$^3$H]-2-DG during an intraperitoneal GTT as described previously[46,47], with minor modifications. Mice that had been denied access to food overnight

received oral administration of 30 mg kg$^{-1}$ of DS20060511 or vehicle 15 min prior to the glucose load. The mice then received intraperitoneal glucose administration (1 g kg$^{-1}$ containing 100 μCi kg$^{-1}$ [$^3$H]-2-DG as tracer), followed by quick removal of the tissues 60 min later. Tissue samples were homogenized in 0.5% perchloric acid and centrifuged, and the supernatants were neutralized with KOH. One aliquot of the supernatants was used for measuring the total radioactivity ([$^3$H]-2-DG and [$^3$H]-2-DG 6-phosphate ([$^3$H]-2-DGP)). A second aliquot of the supernatants was treated with 1 N Ba(OH)$_2$ and 1 N ZnSO$_4$ to remove [$^3$H]-2-DGP, and the [$^3$H]-2-DG count was measured. 2-DG uptake into the tissue, which is rapidly metabolized to 2-DGP in the tissue, was estimated by subtracting the count of [$^3$H]-2-DG from the total count. $^3$H-specific activities were counted using a liquid scintillation counter.

**2-DG uptake in isolated skeletal muscle**. The soleus or EDL muscle was removed from anesthetized mice (isoflurane anesthesia) that had been denied access to food overnight, and incubated with oxygenated (95% O$_2$/5% CO$_2$) Krebs-Henseleit Bicarbonate (KHB) buffer (118.1 mM NaCl, 4.7 mM KCl, 1.1 mM KH$_2$PO$_4$, 1.2 mM MgSO$_4$, 2.5 mM CaCl$_2$, and 25 mM NaHCO$_3$, pH7.4) in the presence of 11.1 mM glucose at 37 °C. The muscle specimens were then treated with the indicated concentrations of the compounds in KHB buffer containing 11.1 mM glucose and 8 mM mannitol for 10 min at 37 °C. Thereafter, the muscles were rinsed with KHB buffer containing 8 mM mannitol and the compounds for 5 min at 37 °C. Lastly, the muscles were treated with the compounds in KHB buffer containing 1 mM 2-DG (with 0.3 μCi mL$^{-1}$ [$^3$H]-2-DG) and 8 mM mannitol (with 0.03 μCi mL$^{-1}$ [$^{14}$C]-mannitol) for 10 min at 37 °C. After the muscle specimens were washed, they were lysed with 1 N NaOH and neutralized with 1 N HCl. The $^{14}$C- and $^3$H-specific activities were counted with a liquid scintillation counter. The specific uptake of 2-DG was calculated by subtracting the nonspecific uptake of mannitol from the total 2-DG uptake. To investigate the effects of muscle contraction, muscle contraction was induced by electrical stimulation with 5 Hz (1 ms pulse duration, 100 V) for 10 min (SEN-5201, Nihon Koden, Tokyo, Japan).

**Plasma membrane fractionation of the skeletal muscle**. Mice that had been denied access to food overnight received DS20060511 (10 mg kg$^{-1}$), insulin (5 U kg$^{-1}$), or saline (vehicle control) via the inferior vena cava under isoflurane anesthesia, and 10 min later, the triceps surae muscle was excised. The plasma membrane fraction of each skeletal muscle specimen was prepared as described previously[48,49]. Briefly, the triceps surae muscle was homogenized in Buffer A (20 mM HEPES, 1 mM EDTA, 1 mM PMSF, and protease inhibitor) containing 250 mM sucrose on ice. The muscle homogenate was centrifuged at 2000 × g for 10 min at 4 °C to remove any unhomogenized muscle fibers. The supernatant was then centrifuged at 19,000 × g for 20 min at 4 °C. The pellet was resuspended in 3 mL Buffer A, layered on a 6 mL sucrose cushion (38% sucrose in Buffer A) and centrifuged at 100,000 × g for 60 min at 4 °C. The membrane fraction recovered on top of the sucrose cushion was resuspended in Buffer A and centrifuged at 40,000 × g for 20 min at 4 °C. The pellet was designated as the plasma membrane fraction and subjected to immunoblotting.

**Sample preparation for immunoassays of insulin- and AMPK-signaling molecules**. For the analyses of insulin and AMPK signal transduction, mice that had been denied access to food overnight received DS20060511 (10 mg kg$^{-1}$), insulin (5 U kg$^{-1}$), or saline (vehicle control) via the inferior vena cava under isoflurane anesthesia and 10 min later, the triceps surae muscle specimen was excised for western blot analysis. The tissue sample was homogenized with a Polytron homogenizer and lysed in a lysis buffer (25 mM Tris-HCl, pH 7.4, 100 mM NaF, 50 mM Na$_4$P$_2$O$_7$, 10 mM EGTA, 10 mM EDTA, 10 mM Na$_3$VO$_4$, 1 mM PMSF, and 1% NP-40) on ice, and the lysate was centrifuged at 17,860 × g for 10 min at 4 °C. The supernatant was collected and the protein concentrations were determined by BCA assay (Thermo Fisher Scientific). Immunoprecipitation of IRβ and IRS1 was performed as described previously[10], using specific antibodies against IRβ (sc-711, Santa Cruz Biotechnology, Dallas, TX), IRS1 (06-248, Merck Millipore, Burlington, MA), and for detection of phosphotyrosine (05-321, Merck Millipore). Five milligrams of the extracts were incubated with specific antibodies against IRS1 for 1 h at 4 °C. Then, protein G-Sepharose was added, followed by incubation for 2 h at 4 °C. After washing three times, the immunocomplexes were subjected to immunoblotting.

**Immunoblotting**. Phosphorylated or total protein of IRβ and IRS1 was resolved on 7% SDS-PAGE and analyzed using specific antibodies against IRβ, IRS1, and phosphotyrosine. Analyses were also conducted for phosphorylated Akt (1:2000), Akt (1:2,000), AS160 (1:1,000), phospho-AS160 (1:1,000), AMPKα (1:1,000), phospho-AMPKα (1:1,000), GLUT4 (1:200), Na,K-ATPaseα (1:1,000), and GAPDH (1:1,000) by immunoblotting with specific antibodies after the tissue lysates were resolved on SDS-PAGE and transferred to a PVDF membrane using Trans-Blot Turbo transfer system (Bio-Rad Laboratories, Hercules, CA). Bound antibodies were detected with HRP-conjugated secondary antibodies using ECL detection reagents (GE Healthcare, Chicago, IL).

**Fluorescence immunostaining**. Differentiated L6-GLUT4myc myotubes were prepared in a collagen-coated four-well chamber slide. After serum starvation for 3 h, the cells were stimulated with insulin or DS20060511 for 15 min at 37 °C. Cells were rinsed with cold PBS(+), fixed with 4% paraformaldehyde for 30 min on ice and blocked with 1% BSA and 10% normal goat serum in PBS(+) for 30 min at room temperature. Surface GLUT4myc staining was carried out without cell membrane permeabilization. Cells were incubated with anti-c-Myc antibody for 30 min at room temperature, followed by treatment with 0.1% Triton-X. After blocking, the cells were incubated with Alexa 488-conjugated secondary antibody and Alexa 594-conjugated phalloidin for 30 min at room temperature. Fluorescence images were obtained with a Leica TSC-SP8 confocal microscope. Specimens were scanned along the z-axis and a single-composite image was generated by the maximal projection method using the LAS software (Leica Microsystems, Wetzlar, Germany).

**Indirect calorimetry under treadmill running**. The mice were acclimatized to the treadmill by allowing them to run at 10 m min$^{-1}$ for 10 min on the day before the test. On the day of the test, the mice received oral administration of 30 mg kg$^{-1}$ of DS20060511 or vehicle 15 min before they were made to run. The treadmill was started at a velocity of 10 m min$^{-1}$, with the speed increased stepwise by 2 m min$^{-1}$ every 3 min, to assess the exercise endurance capacity as described previously[50]. Measurements of the oxygen consumption (VO$_2$) and exhaled carbon dioxide (VCO$_2$) under treadmill running were conducted with the ARCO-2000 magnetic-type mass spectrometric calorimeter (ARCO System, Kashiwa, Japan) connected to the treadmill chamber (MB-2000, ARCO System). The RER was calculated as the VCO$_2$/VO$_2$ ratio and the substrate utilization rates were calculated using Frayn's formula[51]: rate of glucose oxidation = 4.585VCO$_2$ − 3.226VO$_2$; rate of fat oxidation = 1.695VO$_2$ − 1.701VCO$_2$. The exercise endurance capacity under treadmill running was determined by measuring the time from the start to the time until the mice ceased to run because of exhaustion, which was defined as remaining on the shocker plate for more than 10 sec. Immediately after the exercise, the blood glucose and lactate levels were measured using Glutest Every and Lactate Pro2 (ARKRAY, Kyoto, Japan), respectively, in blood samples collected from the tail vein.

**Repeated DS20060511 treatment of the *db/db* mice**. *db/db* mice were acclimatized to a 6 h restricted feeding pattern from 10 am to 4 pm prior to the start of the experiment. For 28 days from 6 weeks of age, the mice received oral administration of 10 mg kg$^{-1}$ of DS20060511 or vehicle once a day, 15 min prior to their feeding. On day 1 and day 28, the blood glucose levels were measured between 10 am to 4 pm. Blood glucose levels were measured using Glutest Every in blood samples collected from the tail vein. Daily food intake was monitored throughout the study, and the body weight, blood glucose, and plasma insulin were measured on day 1 and day 28. The hemoglobin A1c (HbA1c) level was also measured with a HLC-723G8 Automated Glycohemoglobin Analyzer on day 1 and day 28 (Tosoh, Tokyo, Japan). The triceps surae muscle (skeletal muscle), heart, epididymal WAT, and liver were excised immediately after euthanasia, and the tissue weights were measured on day 29. The glycogen contents of the muscle, heart, and liver were measured as described previously,[52] with some modifications. In brief, each tissue sample (50–100 mg) was weighed and boiled in 0.75 mL of 8 M KOH. After cooling, 0.25 mL of 1 M Na$_2$SO$_4$, followed by 2 mL of 99.5% ethanol, was added to precipitate the glycogen. The samples were centrifuged and the glycogen precipitates were dissolved in distilled water. Glycogen was degraded to glucose by treatment with amyloglucosidase (Sigma–Aldrich) at 55 °C for 60 min, and the glucose concentration was determined with Glucose CII Test Wako (FUJIFILM Wako Pure Chemical).

**Pharmacokinetics and metabolic stability of DS20060511**. C57BL/6 mice that had been denied access to food overnight received the indicated oral dose of DS20060511, and blood samples were collected from the tail vein at the indicated timepoints. Plasma samples for quantitation of DS20060511 were prepared by centrifugation. Another group of mice received oral administration DS20060511 (30 mg kg$^{-1}$) and euthanized 75 min later. The soleus and gastrocnemius muscles, heart, and inguinal WAT were removed and homogenized in water (5 vol.) for quantitation of the drug concentrations. Plasma samples and tissue homogenates were analyzed by LC-MS/MS (Nexera MP System, SHIMADZU, Kyoto, Japan; API4000 System, SCIEX, Framingham, MA). The tissue concentrations of the compound were calculated by assuming the volume of 1 g of tissue is 1 mL. Evaluation of the metabolic stability of the compound to mouse and human cytochrome P450 (CYP) was conducted as follows. The test compound (1 μM) was incubated with pooled liver microsomes (0.1 mg mL$^{-1}$) in sodium phosphate buffer (100 mM, pH 7.4) at 37 °C with an NADPH generating system: β-nicotinamide adenine dinucleotide phosphate (β-NADP) 1 mM, D-glucose 6-phosphate (G-6-P) 10 mM, glucose 6-phosphate dehydrogenase (G-6-PDH) 1 U mL$^{-1}$, and magnesium chloride 3.3 mM. After 30 min incubation, the reaction was terminated by addition of acetonitrile and samples were analyzed by LC-MS/MS. The results were expressed as the amount of compound remaining (expressed as a percentage), based on the ratio of the relative area to that in the control (without reaction).

**Antibodies**. Antibodies against c-Myc (9E10, sc-40 and A-14, sc-789), Glut4 (C-20, sc-1608), and IRβ (C-19, sc-711) were purchased from Santa Cruz Biotechnology.

Antibodies against Akt (9272), phospho-Akt (9271), AMPKα (2532), phospho-AMPKα (2531), Na,K-ATPaseα (3010), GAPDH (2118), anti-mouse IgG (7076), and anti-rabbit IgG (7074) were purchased from Cell Signaling Technology, Danvers, MA. Rabbit polyclonal antibody against IRS1 (06-248), mouse monoclonal antibody against phosphotyrosine (05-321), rabbit antiserum against AS160 (07-741), and rabbit polyclonal antibody against phospho-AS160 (07-802) were purchased from Merck Millipore. HRP-conjugated donkey anti-goat IgG (V8051) was purchased from Promega Corporation, Madison, WI. HRP-conjugated goat anti-rabbit IgG (111-035-144) and goat anti-mouse IgG (115-035-003) were purchased from Jackson ImmunoResearch, Laboratories, West Grove, PA.

**Statistics and reproducibility**. Values are expressed as the means ± SEM. Differences between two groups were assessed using Student's $t$-tests and log-rank test was performed for evaluation of the exercise endurance capacity exercise capacity test. Statistically significant differences among multiple groups were evaluated by one-way ANOVA followed by the indicated post-hoc multiple comparisons (Dunnett's test, Williams' test, and Tukey's test). All our results were obtained from distinct samples or independent experiments. Details of each statistical test are indicated in the figure legends. Non-numeric data are shown as representative results from more than three independent experiments.

**Reporting summary**. Further information on research design is available in the Nature Research Reporting Summary linked to this article.

## Data availability

All data generated or analyzed in this this study are available within the article, its Supplementary Information file, and its Supplementary Data 1, or will be made available by the corresponding author upon request.

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

## Acknowledgements
We appreciate the scientific advice provided to us from Drs. Amira Klip and Philip Bilan. We are grateful to Yumi Kawase for her technical assistance. We are also thankful to Drs. Kazushi Kubota, Jun Tanaka, Kazushi Araki, Futoshi Nara, and Masaaki Takahashi for their generous support to the program. This work was supported by a grant for TSBMI from the Ministry of Education, Culture, Sports, Science and Technology of Japan, a Grant-in-Aid for Scientific Research (B) (18H02860), and a grant (B) (15H04847) from the Ministry of Education, Culture, Sports, Science, and Technology of Japan (to N.K.).

## Author contributions
S.F., J.T., M.K., Te.K., M.T., N.K., and T.K. conceived and designed the experiments; S.F., J.T., and A.N. performed the experiments; S.F. and T.J. analyzed the data; J.T., Te.K., and H.K. wrote the manuscript; Te.K., N.K., and T.K. reviewed and revised the manuscript.

## Competing interests
The authors declare the following competing interests: S.F., J.T., M.K., A.N., and H.K. are employees of Daiichi Sankyo Co., Ltd. (Tokyo, Japan). The remaining authors declare no competing interests.
