## [Peer Review File · Communications Biology]

Reviewers' comments:

Reviewer #2 (Remarks to the Author):

Title: A novel xanthene derivative, DS20060511, attenuates glucose intolerance by inducing skeletal muscle-specific GLUT4 translocation in diabetic mice

The author has extensively studied the possible mechanism for DS20060511 and the study showed the mechanism is independent of Insulin, AMPK and NO pathway and assumed that "DS2006051 may promote exocytosis or suppress endocytosis of GLUT4 via target molecule activation". Although the study is novel and investigated extensively, it is lacking critical information for this molecule which is commented as below.

Major comments

1. On page-4, last paragraph: The justification of using the xanthene derivatives need to be more clear. As per the paragraph, the authors have randomly used the chemicals for screening its effect on GLUT4 translocation using myotubes. However, I would suggest you explain more rationally about selecting the xanthene derivatives for this study.
 2. On page-4, Line-79-88, It seems the author has discussed the result in this paragraph. This information would be more suitable for the discussion section.
 3. How did you select the doses of DS20060511 for In vitro and In vivo study?
 4. Do you have Pharmacokinetic, metabolic stability data for in vivo study to support your dose that is administered orally?
 5. Similarly, is the compound distributing to the skeletal muscle with enough quantity to produce the effect? Is there any tissue distribution study?
 6. Do you have any MTT data?
 7. On page-11, Line-214 and 215, Your molecule is intended to be specific for type-2 DM? Why not in Type-1? You have used a type-1 DM animal model i.e. STZ-induced diabetes and DS20060511 has shown reduced blood glucose level along with insulin (Figure-4g).
 8. How did you standardize the treadmill experiment? Is there any reference to support your procedure because treadmill exercise can also utilize fatty acids but you have mentioned that the glucose is only utilized? How did you standardized the method to justify the glucose utilization?
 9. In the discussion, please explain Why the DS2006051 treatment showed an increase in glucose uptake during OGTT in normal mice? Why the insulin secretion was decreased during OGTT in presence of DS2006051? This suggests that DS2006051 increases the GLUT4 translocation in a normal mouse in presence of glucose load.
 10. In GLUT4 knockout mice, why the insulin concentration is higher as compared to WT mice? - Figure-3a
 11. The mechanism of DS2006051 is independent of Insulin, AMPK and NO pathway and the author lastly assumed that DS2006051 showed these effects by promoting exocytosis or suppressing endocytosis of GLUT4.
- The paper will be scientifically superior if the author could provide evidence for his assumption.

Minor Comments

1. Page-2, Line-31: "L6-miytubes"- Correct the spelling
2. Page-4; Line-67: "was not inhibit" correct the grammar
3. What is the High-fat diet used in the study? %Kcal? Source?
4. Page-13, Line:276-277, Please provide the approval number
5. Page-13, Line:282, Provide the mode/instruments used for measuring fasting blood glucose.
6. Source of Streptozotocin? All the reagents, kits and instruments mentioned in the methodology section should have the source with the country of origin.
7. Page-15, Line:327, provide the dose of glucose load
8. Page-17, Line-383, provide the homogenization procedure of skeletal muscle
9. Statistical analysis: you should also mention Tukey's test as a Post-hoc test because you have used this test in some results (Figure-7 legend)

Reviewer #3 (Remarks to the Author):

In the present investigation, Furuzono et al., identify DS20060511, a skeletal muscle cell-specific GLUT4 translocation enhancer. DS20060511 has remarkable effects on lowering blood glucose and enhancing skeletal muscle glucose uptake via GLUT4. It is an interesting study with robust methodology and transgenic models to test mechanisms. As muscle glucose uptake is dysregulated in conditions of obesity, diabetes, and cancer, the findings would be of considerable interest to a broad readership interested in such conditions. It is very interesting that DS20060511 is specific to skeletal muscle cells and it is appreciated that the authors have tried to determine the mechanism of action.

Yet, I have some comments and suggestions for improving the study context, the interpretation of the data, as well as suggestions for a few additional analyses, which I believe will enhance the impact and merit of the study.

Generally, the references are not up to date on the current literature; several old reviews are cited instead of more recent ones (or even better, citing the original paper). Please see below for some suggestions for better placing the manuscript findings into context of the present knowledge in the field.

Line 56-57, it would be better to reference the original studies than an old review (PMID: 17259384)

59-60: Again the references to reviews seem a bit out dated. If ref to a review, please provide a newer review (PMID: 27739515), or refer to an original article.

63: AMPK is well-established not to be implicated in exercise-stimulated glucose uptake (PMID: 32017593 for an update). Rather, NOX2-dependent ROS production induced by Rac1 is implicated (PMIDs: 31604916, 31846370, 28389470, 27061726), which should be acknowledged here instead of the old view on AMPK. Also the case in line 172-173: Please provide the reader with the updated data on AMPK's role in exercise-stimulated glucose uptake.

L64-65: "Furthermore, AICAR, which induces AMPK activation, was shown to increase glucose uptake by the skeletal muscle." Reference is missing, please insert.

71-72: "In subjects with type 2 diabetes, skeletal muscle biopsy specimens obtained during euglycemic insulin clamp showed impaired insulin signaling in the skeletal muscle²⁵." Several studies show no defect in insulin signaling (PMID: 19690335, 10491408) this should be stated to put the study into the correct perspective.

137: "The insulin-induced GLUT4 translocation are activated by 1) the IR-IRS1-PI3K-Akt-AS160 138 pathway, and 2) the IR-IRS1-PI3K-Rac1 pathway in the skeletal muscle." Please insert relevant references for this (incl: 16804077 for 1, and PMID: 23349504 and 17259384 for 2). What is the fate of the glucose in response to DS20060511? Is it oxidized or stored as glycogen? Does it produce excessive oxidative stress? Please include such important measures.

How does the GLUT4 vesicles translocate if there is no actin cytoskeleton? Or is it possible that GLUT4 transporter activity is regulated by DS20060511?

Another target for Rac1 is NOX2 producing ROS at the PM, how was ROS induced by DS20060511. The latrunculin B results does not really rule out the Rac1 signal to glucose uptake, which is likely mediated via NOX2-dependent ROS. The results for Rac1 could be strengthened by testing this DS20060511 in muscle +/- a Rac1 inhibitor or KO.

One argument has been that insulin resistance is a mechanism that set in to protect the muscle from excessive glucose to avoid oxidative stress. Could the authors provide measures of oxidative stress following the 28 days treatment period? Also a more comprehensive characterization of whether increasing glucose uptake to skeletal muscle, without the metabolic demand, might in fact be detrimental to skeletal muscle is needed in order to evaluate if this is a feasible therapeutic treatment strategy.

What was blood glucose levels during exercise and was exercise capacity affected by DS20060511? It could be potentially harmful due to hypoglycemia events during exercise. How is the safety of this drug assessed in vivo?

204: "in obese diabetic mice". These mice were not diabetic, merely DIO and insulin resistant, please correct.

It would be highly informative to measure substrate utilization and activity levels in response to

DS20060511 in free living animals.

It would be beneficial could the authors provide a more robust assessment of the metabolic changes that occurred in the dbdb mice treated for 28 days with DS20060511. How did plasms fatty acids change, inflammation, was hepatic steatosis improved?

Methods:

Please specify the macronutrient composition in the diets

Please specify under which temperature the mice were housed

What temperature was the isolated skeletal muscle incubated at?

Please provide information about lysis buffer

Line 381: "10 mg/body), insulin (5 unit/body) or saline" what is means by unit/body? Please specify.

Please include information about any prior acclimatization to the treadmills before the indirect calorimetry under treadmill running.

Point-by-point response to the reviewer's comments

In response to Reviewer #2

The author has extensively studied the possible mechanism for DS20060511 and the study showed the mechanism is independent of Insulin, AMPK and NO pathway and assumed that DS2006051 may promote exocytosis or suppress endocytosis of GLUT4 via target molecule activation. Although the study is novel and investigated extensively, it is lacking critical information for this molecule which is commented as below.

Response: We are extremely grateful to Reviewer #2 for the very careful review of our manuscript and for the suggestions that we believe, were instrumental in improving the quality of our manuscript.

Major comments

Reviewer #2: Major Comment 1

On page-4, last paragraph: The justification of using the xanthene derivatives need to be more clear. As per the paragraph, the authors have randomly used the chemicals for screening its effect on GLUT4 translocation using myotubes. However, I would suggest you explain more rationally about selecting the xanthene derivatives for this study.

Response: We agree that our description of why we selected xanthene derivatives for this study was insufficient and thank the reviewer for this comment. In the revised manuscript, we have described, in detail, the flow to identification of DS20060511 as the study compound in the first paragraph of the Results section (lines 94–101) and Supplementary Fig. 1. The revised text is as follows: “We screened our chemical library, composed of more than 100,000 compounds, using L6-GLUT4myc myotubes, to identify compounds that would induce translocation of GLUT4 to the cell surface. Two completely different compounds were identified and both passed the counter assay to exclude compounds that would exert toxic effects, such as respiratory chain inhibition. Further in vitro assays revealed that one of the two compounds affected the Akt pathway, so that we finally selected the other, an original xanthene compound, as the hit compound with the potential effect of inducing GLUT4 translocation. Lead optimization of the hit compound finally yielded the more potent xanthene compound, DS20060511 (Fig. 1a and Supplementary Fig. 1).”

Reviewer #2: Major Comment 2

On page-4, Line-79-88, It seems the author has discussed the result in this paragraph. This information would be more suitable for the discussion section.

Response: We agree that this information is not suitable for the Introduction section and thank the

reviewer for this comment. In the revised manuscript, we have revised the last paragraph of the Introduction section, as follows (lines 83–91): “In the present study, we showed that the novel xanthene derivative DS20060511 induced skeletal muscle-specific GLUT4 translocation, independent of the actions of insulin. We used L6 myotubes expressing myc-tagged GLUT4 (L6-GLUT4myc) to screen our chemical compound library, and measured GLUT4 translocation to the cell surface by quantitative anti-myc immunoassay. The effects of the compound on the glucose uptake and whole-body glucose metabolism were examined in a series of in vitro and in vivo experiments. The mechanism of action of the compound was explored by investigating known signaling pathways involved in GLUT4 translocation induced by insulin and exercise. Finally, we evaluated the therapeutic potential of the compound in an obese and insulin-resistant mouse model of type 2 diabetes.”

Reviewer #2: Major Comment 3

How did you select the doses of DS20060511 for In vitro and In vivo study?

Response: We thank the reviewer for this question. In the revised manuscript, we have described the concentration- and dose-finding preliminary experiments for selecting the concentration/dose of DS20060511 for the in vitro and in vivo studies under “**Chemicals**” in the Methods section of the revised manuscript, as follows (lines 303–307): “The concentrations and doses of DS20060511 used for the in vitro and in vivo experiments in this study were selected based on the results of preliminary concentration- and dose-finding experiments, including the GLUT4 translocation assay in L6-GLUT4myc myotubes and GTT, respectively.”

Reviewer #2: Major Comment 4

Do you have Pharmacokinetic, metabolic stability data for in vivo study to support your dose that is administered orally?

Reviewer #2: Major Comment 5

Similarly, is the compound distributing to the skeletal muscle with enough quantity to produce the effect? Is there any tissue distribution study?

Response: We agree that all of this information is very important to provide in support of the doses we used for the in vivo study and thank the reviewer for this comment. In the revised manuscript, we have described the pharmacokinetic evaluations of DS20060511 in the Results section and Supplementary Figs. 2a and b, as follows (lines 131–140): “Changes in the plasma concentration and distribution of DS20060511 to possible target organs/tissues were examined in normal mice. The levels of systemic exposure to DS20060511 after oral administration of the compound was dose-dependent, and the maximal concentrations at 30 min after administration of 1, 10, and 30 mg kg⁻¹ were 0.6, 16.5, and 71.4 μM, respectively (Supplementary Fig. 2a). Measurement of the

DS20060511 concentrations in tissues at 75 min after oral administration (30 mg kg⁻¹) revealed almost comparable concentrations among the skeletal muscle, WAT, and heart (Supplementary Fig. 2b). Consistent with its stable pharmacokinetic profile, the metabolic stability of the compound in the liver microsomal fraction was high (89% and 79% compound remaining after 1-h incubation with the mouse and human liver microsomal fraction, respectively).”

Reviewer #2: Major Comment 6

Do you have any MTT data?

Response: We thank the reviewer for this comment. Regrettably, we do not have any meal tolerance test (MTT) data, however, we evaluated the glucose-lowering effect of DS20060511 after refeeding in *db/db* mice (Figs. 8c and d). The results suggested that DS20060511 treatment improves glucose intolerance, not only after a glucose load, but also after meals.

Reviewer #2: Major Comment 7

On page-11, Line-214 and 215, Your molecule is intended to be specific for type-2 DM? Why not in Type-1? You have used a type-1 DM animal model i.e. STZ-induced diabetes and DS20060511 has shown reduced blood glucose level along with insulin (Figure-4g).

Response: We agree with your comment and thank the reviewer for it. We were originally exploring enhancers of GLUT4 translocation for patients with type 2 diabetes, because glucose uptake into the skeletal muscle has been reported to be attenuated in subjects with type 2 diabetes, but not those with type 1 diabetes, due mainly to insulin resistance. As you point out, DS20060511 administration in combination with insulin, however, also showed a glucose-lowering effect in a mouse model of STZ-induced diabetes (Fig. 4g), suggesting that it may be also useful for type 1 diabetes. We have rewritten the text in the revised manuscript, as follows (lines 265–267): “Moreover, when administered in combination with insulin, DS20060511 further enhanced glucose uptake into the skeletal muscle in both normal and insulin-resistant mice, and further reduced the blood glucose levels in a mouse model of STZ-induced type 1 diabetes. And, in lines 269–271, Thus, DS20060511 may act as an anti-diabetic agent with an entirely novel mechanism of action in patients with impaired actions of insulin in the skeletal muscle and those with either type 1 or 2 diabetes receiving insulin and/or exercise therapy.”

Reviewer #2: Major Comment 8

How did you standardize the treadmill experiment? Is there any reference to support your procedure because treadmill exercise can also utilize fatty acids but you have mentioned that the glucose is only utilized? How did you standardized the method to justify the glucose utilization?

Response: We apologize for not providing the appropriate reference and describe the procedure of the treadmill exercise. In the revised manuscript, we have added the reference No. 50 (PMID: 22174785, line 471) and described the procedure of the test in the Methods section, as follows (lines 470–477): “The treadmill was started at a velocity of 10 m min⁻¹, with the speed increased stepwise by 2 m min⁻¹ every 3 min, to assess the exercise endurance capacity as described previously⁵⁰. Measurements of the oxygen consumption (VO₂) and exhaled carbon dioxide (VCO₂) under treadmill running were conducted with the ARCO-2000 magnetic-type mass spectrometric calorimeter (ARCO System, Kashiwa, Japan) connected to the treadmill chamber (MB-2000, ARCO System). The RER was calculated as the VCO₂/VO₂ ratio and the substrate utilization rates were calculated using Frayn’s formula⁵¹: rate of glucose oxidation = 4.585VCO₂ – 3.226VO₂; rate of fat oxidation = 1.695VO₂ – 1.701VCO₂.” The exercise protocol for the stepwise treadmill running was selected as described previously, since we have intended to evaluate the balance between glucose and fat oxidation (PMID: 22174785). We have indicated in the revised manuscript that DS2006051 treatment during exercise increased glucose oxidation and decreased fat oxidation (lines 186-189 and Fig. 5b, c).

Reviewer #2: Major Comment 9

In the discussion, please explain Why the DS2006051 treatment showed an increase in glucose uptake during OGTT in normal mice? Why the insulin secretion was decreased during OGTT in presence of DS2006051? This suggests that DS2006051 increases the GLUT4 translocation in a normal mouse in presence of glucose load.

Response: Thank you for your comment. During GTT, pre-treatment with DS2006051 increases GLUT4 translocation in the skeletal muscle and suppresses blood glucose elevation by increasing glucose uptake into the skeletal muscle. Decreased blood glucose escalation would lead to reduced glucose-stimulated insulin secretion. We think that DS2006051 would induce GLUT4 translocation even in the absence of a glucose load. To obtain evidence in support, we have added the data of the blood glucose changes after DS2006051 treatment in the absence of a glucose load in Fig. 2a and b. In mice that had continued access to food, oral administration of DS2006051 significantly reduced the blood glucose levels (Fig. 2a), suggesting that the compound is effective even in the absence of a glucose load. Interestingly, this reduction of blood glucose was not observed in mice that had been denied access to food overnight (Fig. 2b), suggesting that the glucose-lowering effect of the compound is within the range of homeostatic maintenance, associated with a relatively low risk of development of hypoglycemia. In the revised manuscript, we have described the above in the Results section (lines 115–119) and first paragraph of the Discussion section (lines 253–259).

Reviewer #2: Major Comment 10

In GLUT4 knockout mice, why the insulin concentration is higher as compared to WT mice? -

Figure-3a

Response: GLUT4-knockout mice have been reported to have higher plasma insulin concentrations as compared to wild-type mice (PMID: 7675081). This could represent a compensation for the partial impairment of glucose uptake in these mice due to the loss of GLUT4 expression.

Reviewer #2: Major Comment 11

The mechanism of DS2006051 is independent of Insulin, AMPK and NO pathway and the author lastly assumed that DS2006051 showed these effects by promoting exocytosis or suppressing endocytosis of GLUT4. The paper will be scientifically superior if the author could provide evidence for his assumption.

Response: Thank you for this comment and suggestion. We made a vigorous attempt tried to clarify the mechanisms of action of the compound, including to evaluate exocytosis/endocytosis of GLUT4. Unfortunately, however, we have still not obtained robust data. We believe that it is possible that the compound has effects on exocytosis and/or endocytosis of GLUT4. Further investigation is needed to clarify the mechanisms of action of DS20060511 causing cell surface translocation of GLUT4, as well as to identify the molecular target.

Reviewer #2: Minor Comment 1

Page-2, Line-31: L6-miytubes Correct the spelling

Response: Thank you for pointing out the typographical error. In the revised manuscript, we have corrected the spelling to “L6-myotubes” in line 32.

Reviewer #2: Minor Comment 2

Page-4; Line-67: was not inhibit Correct the grammar

Response: Thank you for pointing out the grammatical error. In the revised manuscript, we have corrected the erroneous phrase to “was not inhibited” in line 70.

Reviewer #2: Minor Comment 3

What is the High-fat diet used in the study? %Kcal? Source?

Response: Thank you for your comment. In the revised manuscript, we have added information about high fat diet under “**Animals**” in the Methods section, as follows (lines 321–324): “The HFD32 (CLEA Japan) used as the HFD had a calorie ratio of protein: fat: carbohydrate of 20.1: 56.7: 23.2, with a metabolic calorie content of 5.1 kcal g⁻¹. The sources of the fat were safflower oil and

beef tallow (20.0% and 15.9% in weight, respectively).”

Reviewer #2: Minor Comment 4

Page-13, Line:276-277, Please provide the approval number

Response: Thank you for pointing out the omission. In the revised manuscript, we have added the approval number under “**Animals**” in the Methods section, as follows (lines 325–328): “The animal care and experimental procedures used in the study were approved by The University of Tokyo Animal Care Committee (Approval number: 27-3), and the study was carried out in accordance with the Animal Experimentation Guidelines of Daiichi-Sankyo Co., Ltd (Approved number: 20000411).”

Reviewer #2: Minor Comment 5

Page-13, Line:282, Provide the mode/instruments used for measuring fasting blood glucose.

Response: Thank you for pointing out the omission. In the revised manuscript, we have specified the instrument used for measuring the blood glucose, as follows (line 334): “Blood glucose levels were measured using Glutest Every (Sanwa Kagaku Kenkyusho, Nagoya, Japan) in blood samples collected from the tail vein.”

Reviewer #2: Minor Comment 6

Source of Streptozotocin? All the reagents, kits and instruments mentioned in the methodology section should have the source with the country of origin.

Response: Thank you for your suggestion. In the revised manuscript, we have added the source of streptozotocin in the Methods section (line 309): “Streptozotocin was purchased from FUJIFILM Wako Pure Chemical, Osaka, Japan.” We have also added the sources with the country of origin of the other reagents/instruments.

Reviewer #2: Minor Comment 7

Page-15, Line:327, provide the dose of glucose load

Response: Thank you for your careful review. In the revised manuscript, we have added the dose of glucose load for GTT, as follows (line 376): “Mice that had been denied access to food overnight received the indicated oral dose of DS20060511 or vehicle 15 min prior to the glucose load (1.5 g kg⁻¹, except for eNOS-knockout mice, which received 3.0 g kg⁻¹).”

Reviewer #2: Minor Comment 8

Page-17, Line-383, provide the homogenization procedure of skeletal muscle

Response: Thank you for this suggestion. In the revised manuscript, we have added the homogenization procedure for the skeletal muscle specimens, as follows (line 435): “The tissue sample was homogenized with a Polytron homogenizer and lysed in a lysis buffer (25 mM Tris-HCl, pH 7.4, 100 mM NaF, 50 mM Na₄P₂O₇, 10 mM EGTA, 10 mM EDTA, 10 mM Na₃VO₄, 1 mM PMSF, and 1% NP-40) on ice, and the lysate was centrifuged at 17,860 × g for 10 min at 4°C.”

Reviewer #2: Minor Comment 9

Statistical analysis: you should also mention Tukey’s test as a Post-hoc test because you have used this test in some results (Figure-7 legend)

Response: Thank you for this suggestion. In the revised manuscript, we have added information on the post-hoc tests used in this study in the Methods section, under “**Statistical and reproducibility**,” as follows (line 532): “Statistically significant differences among multiple groups were evaluated by one-way ANOVA followed by the indicated post-hoc multiple comparisons (Dunnett’s test, Williams’ test, and Tukey’s test).”

In response to Reviewer #3

In the present investigation, Furuzono et al., identify DS20060511, a skeletal muscle cell-specific GLUT4 translocation enhancer. DS20060511 has remarkable effects on lowering blood glucose and enhancing skeletal muscle glucose uptake via GLUT4. It is an interesting study with robust methodology and transgenic models to test mechanisms. As muscle glucose uptake is dysregulated in conditions of obesity, diabetes, and cancer, the findings would be of considerable interest to a broad readership interested in such conditions. It is very interesting that DS20060511 is specific to skeletal muscle cells and it is appreciated that the authors have tried to determine the mechanism of action. Yet, I have some comments and suggestions for improving the study context, the interpretation of the data, as well as suggestions for a few additional analyses, which I believe will enhance the impact and merit of the study. Generally, the references are not up to date on the current literature; several old reviews are cited instead of more recent ones (or even better, citing the original paper). Please see below for some suggestions for better placing the manuscript findings into context of the present knowledge in the field.

Response: We are extremely grateful to Reviewer #3 for the very careful review of our manuscript and for the generous suggestions, which we believe were instrumental in improving the quality of our manuscript.

Reviewer #3: Comment 1

Line 56-57, it would be better to reference the original studies than an old review (PMID: 17259384)

Response: Thank you for your suggestion; in the revised manuscript, we have cited the original study (PMID: 17259384), as reference No. 15 (line 57).

Reviewer #3: Comment 2

59-60: Again the references to reviews seem a bit out dated. If ref to a review, please provide a newer review (PMID: 27739515), or refer to an original article.

Response: Thank you for your suggestion; in the revised manuscript, we have cited a newer review (PMID: 27739515), as reference No. 16 (line 61).

Reviewer #3: Comment 3-5

63: AMPK is well-established not to be implicated in exercise-stimulated glucose uptake (PMID: 32017593 for an update). Rather, NOX2-dependent ROS production induced by Rac1 is implicated (PMIDs: 31604916, 31846370, 28389470, 27061726), which should be acknowledged here instead of the old view on AMPK.

Also the case in line 172-173: Please provide the reader with the updated data on AMPKs role in

exercise-stimulated glucose uptake.

L64-65: Furthermore, AICAR, which induces AMPK activation, was shown to increase glucose uptake by the skeletal muscle. Reference is missing, please insert.

Response: Thank you for your great suggestion. In the revised manuscript, we have made major changes in the Introduction section, as follows (lines 64–69): “Despite the reported evidence of contraction inducing phosphorylation of TBC1D1 by activating AMPK¹⁹ or of increased skeletal muscle glucose uptake by pharmacological activation of AMPK by AICAR²⁰, the significance of AMPK in exercise-stimulated glucose uptake in vivo remains controversial^{21,22}. Recently, induction by Rac1 of NADPH oxidase 2-dependent production of reactive oxygen species was implicated in glucose uptake during exercise, through regulation of GLUT4 translocation^{23,24}.” We have also added updated data on the role of AMPK in exercise-stimulated glucose uptake in the Results section, as follows (lines 199–202): “Although recent findings suggest that AMPK plays no role in the GLUT4 translocation and glucose uptake in the muscle observed during exercise^{16,22}, activation of AMPK by electrical stimulation²¹, as well as by AICAR²⁰, could increase the glucose uptake in isolated skeletal muscle.” In addition, we have described NOX2-dependent ROS production at the end of the Discussion section, as follows (lines 292–294): “Further investigation to identify the molecular target of DS20060511 and also the signaling pathway involved, such as Rac1 or NADPH oxidase 2-associated reactive oxygen species production, is needed.”

Reviewer #3: Comment 6

71-72: In subjects with type 2 diabetes, skeletal muscle biopsy specimens obtained during euglycemic insulin clamp showed impaired insulin signaling in the skeletal muscle. Several studies show no defect in insulin signaling (PMID: 19690335, 10491408) this should be stated to put the study into the correct perspective.

Response: Thank you for your suggestion. In the revised manuscript, our reference to the insulin signaling pathway is more precise, and we have added a reference (PMID: 10491408), Reference No. 29 (lines 75–78): “In subjects with type 2 diabetes, skeletal muscle biopsy specimens obtained during a euglycemic insulin clamp showed impaired insulin signaling, observed as reduction in IRS1 phosphorylation and PI3K activity, in the skeletal muscle²⁸, while no effect was noted on the phosphorylation/activity of Akt²⁹.”; however, we think that the reference, PMID: 19690335, may not be suitable for this context, because this is about AMPK signaling in mice.

Reviewer #3: Comment 7

*137: The insulin-induced GLUT4 translocation are activated by 1) the IR-IRS1-PI3K-Akt-AS160
138: pathway, and 2) the IR-IRS1-PI3K-Rac1 pathway in the skeletal muscle. Please insert relevant references for this (incl: 16804077 for 1, and PMID: 23349504 and 17259384 for 2).*

Response: Thank you for your suggestion. In the revised manuscript, we have added references (reference No. 32, PMID: 16804077 for 1 and reference No. 15, PMID: 17259384 for 2) in lines 154–156. We think that the reference, PMID: 23349504, is not suitable for this context, because this is about exercise.

Reviewer #3: Comment 8

What is the fate of the glucose in response to DS20060511? Is it oxidized or stored as glycogen? Does it produce excessive oxidative stress? Please include such important measures.

Response: Thank you for your question. We agree that such information is important. In the revised manuscript, we have added data on the glycogen contents of the skeletal muscle, heart, and liver after 28 days of DS20060511 treatment in *db/db* mice, in Supplementary Fig. 5c and lines 241–243, as follows: “There were also no significant changes in the tissue weights of the muscle, heart, WAT, and liver, or in the glycogen contents of the muscle, heart, and liver (Supplementary Fig. 5b, c).” The finding of no change in the glycogen content of the skeletal muscle suggests that the glucose may be oxidized, and not stored as glycogen. We, however, regret not assessing the level of oxidative stress in the skeletal muscles of the *db/db* mice. To determine whether DS20060511 treatment might produce excessive oxidative stress, the data of indirect calorimetry under GTT in B6 mice may be useful. The figure shown below shows no change in the VO_2 , suggesting that treatment with

DS20060511 does not increase substrate oxidation or produce excessive oxidative stress, at least at the whole-body level in the normal state.

Indirect calorimetry under the oral glucose tolerance test. C57BL/6 mice (n = 6) that had been denied access to food overnight received oral administration of 30 mg/kg DS20060511 or vehicle 15 min prior to the glucose load (1.5 g kg⁻¹). Indirect calorimetry was performed for 1 h after the glucose load.

Reviewer #3: Comment 9

How does the GLUT4 vesicles translocate if there is no actin cytoskeleton? Or is it possible that GLUT4 transporter activity is regulated by DS20060511?

Response: Thank you for this comment. Unfortunately, we failed to clearly elucidate how cell surface translocation of GLUT4 was enhanced by DS20060511 without remodeling of actin. We do

not yet have data to suggest that DS20060511 directly upregulates GLUT4 transporter activity. We think it is possible that DS20060511 induces GLUT4 translocation without actin remodeling, because the actin cytoskeleton exists under the no-insulin and no-exercise conditions. As noted in the Discussion section, exocytosis and/or endocytosis of GLUT4 are possible candidates to explain the mechanism underlying the increased GLUT4 translocation induced by DS20060511. While we attempted to evaluate exocytosis/endocytosis of GLUT4, we, unfortunately, have still not obtained robust data.

Reviewer #3: Comment 10

Another target for Rac1 is NOX2 producing ROS at the PM, how was ROS induced by DS20060511. The latrunculin B results does not really rule out the Rac1 signal to glucose uptake, which is likely mediated via NOX2-dependent ROS. The results for Rac1 could be strengthened by testing this DS20060511 in muscle +/- a Rac1 inhibitor or KO.

Response: We agree with your notion and thank you for your excellent suggestion. In order to completely rule out the involvement of Rac1 signaling, a study using Rac1-KO mice would be required; however, regretfully, it is difficult for us, at present, to perform a study using Rac1-KO mice. Although experiments using Rac1 inhibitors or with measurement of ROS production following DS20060511 treatment could be feasible, we are afraid that such experiments conducted by us without expertise might be inadequate to conclude the involvement of RAC1-mediated NOX2-dependent ROS. We sincerely hope that the issue will be clarified by experts in this field using DS20060511.

Reviewer #3: Comment 11

One argument has been that insulin resistance is a mechanism that set in to protect the muscle from excessive glucose to avoid oxidative stress. Could the authors provide measures of oxidative stress following the 28 days treatment period? Also a more comprehensive characterization of whether increasing glucose uptake to skeletal muscle, without the metabolic demand, might in fact be detrimental to skeletal muscle is needed in order to evaluate if this is a feasible therapeutic treatment strategy.

Response: We agree that such information is important and thank you for your suggestion. Regrettably, we did not assess the level of oxidative stress in the skeletal muscles of the *db/db* mice. So far, we have performed indirect calorimetry under various conditions, including during exercise, GTT, and the normal state, and we have never noted increase of the VO_2 following DS20060511 treatment under any condition. This suggests that DS20060511 does not increase substrate oxidation, at least at the whole-body level. The slight increase in the RER observed during the exercise endurance study (Fig. 5a) may suggest modest shift of the energy substrate from lipids to

carbohydrates, but it was not at a level sufficient to increase the oxidative stress. Also, we did not find any visible sign of toxicity in the 28-day repeated-dose study of DS20060511 in the *db/db* mice, and the mice showed no change of the muscle weight, food intake, or body weight.

Reviewer #3: Comment 12

What was blood glucose levels during exercise and was exercise capacity affected by DS20060511? It could be potentially harmful due to hypoglycemia events during exercise. How is the safety of this drug assessed in vivo?

Response: We agree that this information is important and thank you for your comment. In the revised manuscript, we have added data from the exercise endurance capacity test, as follows (lines 183–185): “During the stepwise treadmill exercise, the VO₂ gradually increased in both the vehicle- and DS20060511-treated groups (Supplementary Fig. 4a), and the exercise endurance capacity was also comparable between the two groups (Supplementary Fig. 4b).” We have also added data on the glucose levels after the treadmill exercise test, as follows (lines 190–192): “The blood glucose levels decreased significantly after exercise in the DS20060511-treated mice, but did not dip to the hypoglycemia range. The blood lactate levels were comparable between the two groups (Supplementary Fig. 4c).” While we did not observe any signs of toxicity in the mice treated with DS20060511 in our study, thorough toxicological evaluation is warranted, if the compound is to be used in humans.

Reviewer #3: Comment 13

204: in obese diabetic mice. These mice were not diabetic, merely DIO and insulin resistant, please correct.

Response: Thank you for pointing out our error. In the revised manuscript, we have changed “obese diabetic mice” to “mice with diet-induced obesity and insulin resistance” (lines 227 and 234).

Reviewer #3: Comment 14

It would be highly informative to measure substrate utilization and activity levels in response to DS20060511 in free living animals.

Response: We agree that measurement of substrate utilization, such as metabolome analysis, would be very useful for understanding the fate of the increased glucose uptake in response to DS20060511 and thank the reviewer for this suggestion. However, we think that metabolome analysis would be the next step for understanding the mechanism of action of DS20060511.

Reviewer #3: Comment 15

It would be beneficial could the authors provide a more robust assessment of the metabolic changes that occurred in the db/db mice treated for 28 days with DS20060511. How did plasms fatty acids change, inflammation, was hepatic steatosis improved?

Response: We agree that these data would be useful and thank the reviewer for this comment, but we regret to state that we did not measure the levels of FFAs, inflammation markers or the degree of hepatic steatosis following the 28-day repeated-dose study of DS20060511 in the *db/db* mice. However, our study did indicate that there was no change of the liver weight following the treatment, suggesting, at least, that DS20060511 treatment did not deteriorate the severity of hepatic steatosis. In the revised manuscript, we have added data about the tissue weights of the muscle, heart, WAT, and liver in Supplementary Fig. 5b and in lines 244–243, as follows: “There were also no significant changes in the tissue weights of the muscle, heart, WAT, and liver, or in the glycogen contents of the muscle, heart, and liver (Supplementary Fig. 5b, c).”

Reviewer #3: Comment 16

Please specify the macronutrient composition in the diets?

Response: Thank you for your comment. In the revised manuscript, we have added information about the macronutrient composition of the diets in the Methods section, as follows (lines 320–324): “The CE2 normal rodent chow (CLEA Japan) had a calorie ratio of protein: fat: carbohydrate of 29.5: 11.9: 58.5, with a metabolic calorie content of 3.4 kcal g⁻¹. The HFD32 (CLEA Japan) used as the HFD had a calorie ratio of protein: fat: carbohydrate of 20.1: 56.7: 23.2, with a metabolic calorie content of 5.1 kcal g⁻¹. The sources of the fat were safflower oil and beef tallow (20.0% and 15.9% in weight, respectively).”

Reviewer #3: Comment 17

Please specify under which temperature the mice were housed?

Response: Thank you for your question. In the revised manuscript, we have added information about the temperature at which the animals were housed in the Methods section, as follows (lines 317–318): “The mice were group-housed under controlled illumination (12:12-h light-dark cycle) and temperature (23°C ± 2°C) conditions and given free access to normal chow and water, unless otherwise specified.”

Reviewer #3: Comment 18

What temperature was the isolated skeletal muscle incubated at?

Response: Thank you for your question. Isolated skeletal muscle specimens were incubated at 37°C.

In the revised manuscript, we have added information about the temperature at which the muscle specimens were incubated in the Methods section (lines 407–411).

Reviewer #3: Comment 19

Please provide information about lysis buffer.

Response: Thank you for your comment. In the revised manuscript, we have added information about the lysis buffer used in the Methods section, as follows (line 411): “After the muscle specimens were washed, they were lysed with 1N NaOH and neutralized with 1N HCl.”; (lines 435–438): “The tissue sample was homogenized with a Polytron homogenizer and lysed in a lysis buffer (25 mM Tris-HCl, pH 7.4, 100 mM NaF, 50 mM Na₄P₂O₇, 10 mM EGTA, 10 mM EDTA, 10 mM Na₃VO₄, 1 mM PMSF, and 1% NP-40) on ice, and the lysate was centrifuged at 17,860 × g for 10 min at 4°C.”

Reviewer #3: Comment 20

Line 381:10 mg/body), insulin (5 unit/body) or saline; what is means by unit/body? Please specify.

Response: Thank you for pointing out our omissions. In the revised manuscript, we have specified this information in the Methods section, as follows (line 432): “mice that had been denied access to food overnight received DS20060511 (10 mg kg⁻¹), insulin (5 U kg⁻¹) or saline (vehicle control).”

Reviewer #3: Comment 21

Please include information about any prior acclimatization to the treadmills before the indirect calorimetry under treadmill running.

Response: Thank you for pointing out our omission. In the revised manuscript, we have added information about prior acclimatization, before the test, of the animals to the treadmills in the Methods section, as follows (line 467): “The mice were acclimatized to the treadmill by allowing them to run at 10 m min⁻¹ for 10 min on the day before the test.”

REVIEWERS' COMMENTS:

The authors have satisfactorily addressed my comments.